# ARG-Mask RCNN: An Infrared Insulator Fault-Detection Network Based on Improved Mask RCNN

**DOI:** 10.3390/s22134720

**Published:** 2022-06-22

**Authors:** Ming Zhou, Jue Wang, Bo Li

**Affiliations:** School of Information Science and Engineering, Dalian Polytechnic University, Dalian 116039, China; 210510803000492@xy.dlpu.edu.cn (M.Z.); libolb@dlpu.edu.cn (B.L.)

**Keywords:** ARG-Mask RCNN, GA-GD algorithm, insulator fault, ResNet101, rotation mechanism, attention mechanism

## Abstract

Traditional power equipment defect-detection relies on manual verification, which places a high demand on the verifier’s experience, as well as a high workload and low efficiency, which can lead to false detection and missed detection. The Mask of the regions with CNN features (Mask RCNN) deep learning model is used to provide a defect-detection approach based on the Mask RCNN of Attention, Rotation, Genetic algorithm (ARG-Mask RCNN), which employs infrared imaging as the data source to assess the features of damaged insulators. For the backbone network of Mask RCNN, the structure of Residual Network 101 (ResNet101) is improved and the attention mechanism is added, which makes the model more alert to small targets and can quickly identify the location of small targets, improve the loss function, integrate the rotation mechanism into the loss function formula, and generate an anchor frame where a rotation angle is used to accurately locate the fault location. The initial hyperparameters of the network are improved, and the Genetic Algorithm Combined with Gradient Descent (GA-GD) algorithm is used to optimize the model hyperparameters, so that the model training results are as close to the global best as possible. The experimental results show that the average accuracy of the insulator fault-detection method proposed in this paper is as high as 98%, and the number of frames per second (FPS) is 5.75, which provides a guarantee of the safe, stable, and reliable operation of our country’s power system.

## 1. Introduction

With the continuous increase in people’s demand for electricity, the scale of transmission lines is also expanding [1]. The geographical environment where the lines pass is complex and changeable and suffers from severe weather and climate all year round [2]. As a bridge between live conductors or between conductors and the ground, insulators play a role in fixing the busbar and live conductors in power transmission. However, insulator faults occur frequently in reality [3]. Globally, more than 75% of power grid accidents are caused by insulator failures every year, which seriously threatens the safe and stable operation of power grids [4,5]. Various scholars have made efforts to create a healthy and sustainable power grid environment and improve the detection accuracy of faulty insulators [6,7]. The current fault-diagnosis methods [8] can be divided into two camps; one is the physical method, and the other is the method based on deep learning.

As a traditional diagnostic method, physical methods [9] have the advantages of being real-time and high-precision, mainly including ultrasonic, ultraviolet pulse, terahertz, and other methods. Deng et al. [10] proposed an ultrasonic-based insulator peeling detection method. The authors analyzed the propagation speed and energy attenuation of longitudinal and torsional ultrasonic waves in the insulator double-layer model and conducted experiments on the debonding of composite insulators. The results show that the location of the peeling defect can be accurately determined by detecting the propagation attenuation coefficient of the waveguide. Ji et al. [11] proposed a method for detecting the contamination state of ceramic insulators based on ultraviolet pulses. The authors analyzed the results of experiments and operations by monitoring the insulator strings under 110 KV transmission lines in real time. Online monitoring of the insulators is carried out to effectively avoid the occurrence of flashover accidents. Cheng et al. [12] proposed an aging detection of silicone rubber composite insulators based on terahertz technology, using a terahertz vector network analyzer to test the calibrated groups of samples, and an aging detection model of composite insulators based on terahertz signal transmission characteristics is established. However, the above methods are difficult to achieve large-area outdoor detection, and the efficiency is low and requires a large number of professionals to complete. Physical methods are difficult to meet the basic requirements of power grid equipment maintenance.

In recent years, with the continuous development of artificial intelligence technology, detection methods based on deep learning frameworks have been widely used [13]. The method of using drones to photograph and inspect can meet the requirements of large-area outdoor fault detection and improve the efficiency of fault detection [14]. A large number of target-detection algorithms have been applied to insulator fault detection. For example, cascade of the regions with cnn features (Cascade RCNN) [15], single shot multi-box detector (SSD) [16], RetinaNet [17], Mask RCNN [18], you only look once (YOLO) [19] and other methods. Liu et al. [20] proposed an improved SSD insulator-detection algorithm, using a lightweight network MnasNet [21] as a feature extraction network, and then using a multi-scale fusion method to fuse the feature maps. The author used the dataset of aerial images to conduct experiments. The results show that the algorithm can effectively detect the position of the insulator and has the advantages of small model size and fast detection speed. Wen et al. [22] proposed a Cascade RCNN insulator defect-detection method, proposed an algorithm that integrates a series of advanced structures of FPN, cascade regression, and GIoU, and introduced RoI Align instead of RoI pooling to solve the dislocation problem, and introduced depthwise separable convolution and linear bottleneck to reduce the computational burden; the results show that this method can effectively detect defective insulators. Liu et al. [23] proposed an improved RetinaNet-based defect insulator-detection algorithm, which corrected the shortcomings of the Apriori-based RetinaNet anchor box extraction mechanism and used the improved K-means++ algorithm [24] to redesign the number and size of anchor boxes, construct a feature pyramid based on DenseNet as the backbone network, and the experimental results show that this method has obvious advantages in the detection accuracy of insulator defects. Liu et al. [25] proposed an improved YOLO tiny (MTI-YOLO) insulator-detection algorithm, which uses a multi-scale fusion and spatial pyramid pooling (SSP) model and verified the results by comparing with YOLO tiny and YOLO v2. The average accuracy of the proposed algorithm is significantly higher than the above two algorithms, and it can achieve good performance under the condition of complex background and high exposure. The above algorithm belongs to the object-detection algorithm whose output is in the form of a bounding box. For large targets such as insulators, the algorithm can complete the positioning task. However, it is obviously difficult to accomplish multi-type fault identification. The reasons are as follows: as we all know, the insulator failure caused by cracks takes up such a small area. If we continue to use this algorithm to generate anchor frames, we can only determine the approximate location of cracks. However, as a segmentation algorithm, Mask RCNN can accurately detect that the edge positions of the cracks are segmented. Wang et al. [26] proposed a fault-diagnosis method for infrared insulators based on Mask RCNN, using the Mask RCNN network to automatically extract multiple insulators, and using transfer learning and dynamic learning rate algorithms to train the dataset. The experimental results show that the model has high recognition accuracy and calculation speed.

In general, these existing advanced insulator fault-diagnosis methods have their advantages, but some flaws are hard to hide. Physical methods such as ultrasound, ultraviolet pulse, and terahertz, it has the advantages of real-time and high precision. However, they also have common shortcomings, such as it is difficult to achieve large-area outdoor detection, and the efficiency is relatively low. For SSD, RetinaNet, YOLO, Cascade RCNN, and Mask RCNN, these deep learning-based methods have high efficiency and can meet the needs of large-scale outdoor detection, but they have contradictions in real-time and accuracy. Specifically, single-stage target-detection algorithms such as SSD, RetinaNet, and YOLO have fast recognition speed, but low accuracy. The two-stage target-detection algorithms such as Cascade RCNN and Mask RCNN are characterized by high accuracy, but slow speed and difficult to realize real-time monitoring of insulators. It is worth noting that these deep learning-based methods only detect a single fault type, and they cannot complete the multi-fault classification task.

To complete the detection of various faults under the premise of real-time and high precision. In this paper, a fault-diagnosis method for infrared insulators based on ARG-Mask RCNN is proposed. First, it is proposed to modify the 7 × 7 convolution kernel of the first layer of the backbone network ResNet101 to a three-layer 3 × 3 convolution kernel. The three-layer 3 × 3 convolution kernel has the same receptive field as the 7 × 7 large convolution kernel. However, the amount of computation is much smaller than that of the large convolution kernel, and an attention mechanism is added to reduce the amount of network computation and improve the detection speed of small targets. Subsequently, a rotation mechanism is added to the calculation formula of the improved loss function to improve the positioning accuracy of the target insulator and effectively separate the target from the background. After that, it is proposed to improve the initial parameters, and the updated parameters originally generated by Mask RCNN are now generated by a genetic algorithm, to obtain the global optimal solution and improve the identification accuracy of faulty insulators. Then, the labeled dataset is trained to analyze various misdiagnosis phenomena and their causes in the detection results. Finally, the ARG-Mask RCNN method proposed in this paper has obvious advantages through application experiments and comparative analysis. This research has the following contributions:(1)A new backbone network is proposed to improve the capability of fault feature extraction.(2)A rotated anchor box is proposed to reduce the extraneous background in the prediction box.(3)The genetic algorithm combined with the gradient descent method is proposed to optimize the parameters so that the model is as close to the global optimal solution as possible, and the detection accuracy of the model is improved.(4)By comparing with several optimal insulator fault-identification algorithms, the superiority of the proposed method is confirmed.

The rest of this article is organized in the following way. Section 2 briefly introduces the four most common insulator faults and the Mask RCNN base network. Section 3 introduces the ARG-Mask RCNN network in detail from three aspects: backbone network, loss function, and parameter optimization. Section 4 mainly demonstrates the superiority of this method in practical detection. The conclusion is in Section 5.

## 2. Related Work

The infrared data set can reflect the temperature change of each part of the insulator equipment, and the fault detection of the insulator can be carried out according to the thermal imaging results. Mask RCNN network, as a two-stage target-detection algorithm, classifies different faults by continuously learning the characteristics of these faults, and segments the location of faulty insulator strings. This section will explain the data source and Mask RCNN network model.

### 2.1. Data Sources

The infrared data source will be explained below, firstly indicating the characteristics of infrared imaging technology, then introducing the four types of faults with the highest appearance rate, and finally emphasizing the matters needing attention when collecting infrared data. These works will play a crucial role in the labeling of the dataset and later training.

(1)Compared with other fault-diagnosis data, infrared imaging data has the following outstanding characteristics. (I) The data collection is convenient, and the work efficiency is high. It only takes a few hours to complete the collection of a large amount of data with the drone. (II) During the actual inspection, it can be obtained without touching the equipment to avoid product damage caused by improper operation during inspection. (III) A variety of typical faults can be detected, and the location of the faulty insulator sheet and the degree of damage can be located. (IV) Infrared light can detect the internal characteristics of the equipment when it is running. The location of the fault can be identified by the color of the light, which is related to its fault principle, while it is difficult to find faults caused by cracks and internal defects with visible light.(2)To detect a variety of different faults of insulators, it is necessary to determine which type of fault is caused when the data set is marked. The quality of the data set will directly affect the identification of faulty insulators. To avoid confusion and the inability to identify different fault types, the following will introduce the characteristics of four typical infrared faults in detail.

The fault classification of inferior insulators is shown in Figure 1. (I) The type of fault caused by self-explosion can cause some insulator pieces to be missing. (II) Stain and dust fault, common surface stains such as ice and branches will cause the surface temperature of the insulator to exceed 1000 degrees Celsius. (III) Zero-value insulator, the surface of the zero-value insulator fault is dark red. (IV) The insulator sheet is broken, and the temperature difference between the phases of the insulator sheet at the fracture is greater than 18 degrees Celsius.

(3)When collecting data on insulators outdoors, to accurately reflect the temperature of each insulator, the following points should be noted. (I) Weather conditions—avoid collecting in bad weather such as strong wind, strong light, rain, and snow, which will cause the detected device temperature to be inaccurate. (II) The collection time should be selected as early as possible in the morning or the evening when the surface temperature of the insulator is in a relatively stable state. (III) The measurement position should cover the overall map of the insulator string as much as possible. If it is the first measurement, it should keep a certain distance from the equipment to avoid damage to the equipment caused by operation errors.

### 2.2. Mask RCNN Network

The Mask RCNN network [27], first proposed by He et al. in 2017, uses instance segmentation to achieve human pose estimation. Compared with other target-detection algorithms, Mask RCNN generates high-quality pixel-to-pixel masks for each instance, can complete pixel-level segmentation tasks, and has high target-positioning accuracy, which is why this network is selected for insulator-fault detection. This section will describe the Mask RCNN network model in detail, including its backbone network, mask prediction, and region-of-interest correction. At the same time, for the loss function part, the loss function is divided into three parts: mask loss ($L_{mask}$), classification loss ($L_{cls}$), and regression loss ($L_{box}$).

#### 2.2.1. Network Model

Mask RCNN adopts a two-stage network model. In the first stage, Region Proposal Network (RPN) makes predictions on Regions of Interest (ROI). In the second stage, the fully linked network (FCN) predicts the category, offset box, and binary mask of each ROI in parallel. The network model mainly includes the following three parts:Backbone network

In the Mask RCNN model, the ResNet50/101 + FPN model is used as the backbone network. The low-level feature maps have high resolution and weak semantic information, while the high-level feature maps have low resolution and strong semantic information. The higher the resolution, the better for locating small objects, and the stronger the semantic information, the better for classification. They are contradictory. To solve this problem, FPN is proposed as shown in Figure 2, which integrates low-level features and high-level features, that is, it has strong location information and semantic information. The low-level feature information is up-sampled, the feature map gradually becomes larger, and the semantic information is also enhanced. At the same time, the low-level feature maps with strong location information are horizontally connected. FPN enables the network to achieve both precise positioning and strong semantic information.

Pixel Prediction (Mask Prediction)

Mask Prediction is a prediction for pixels; the same pixel value is classified into one category and filled with the same color, and different pixel values are classified and covered by a different color, and pixel-level instance segmentation is conducted.

Region of Interest Align (RoI Align)

RoI Pooling is improved in Mask RCNN. RoI Pooling quantizes a floating-point RoI into the discrete granularity of the feature map, and the quantized RoI is subdivided into spatial containers, which are themselves quantized. In both processes, floating-point numbers are rounded, resulting in the loss of some feature information, which in turn affects the accuracy of the model. To solve this problem, RoI Align is proposed to retain the decimals of the RoI bounding box data, and divide it evenly when subdividing max pooling, retaining the significant digits after the decimal point. When RoI Align performs max pooling, the RoI bounding box can be divided equally, and the center point of each small box can be determined. This point can correspond to four boundary points of the feature map, and bilinear interpolation is performed on these four boundary points. You can determine the value of the center point and then take the maximum value to complete the max-pooling operation.

#### 2.2.2. Loss Function

As one of the important parameters to determine the prediction effect of the deep neural network, the loss function determines the convergence effect of the model to a large extent, and also controls the objective of the network. The smaller the loss function value, the better the performance of the model.

Loss Function: L = $L_{cls}$ + $L_{box}$ + $L_{mask}$

$L_{mask}$ applies only to the true class of the kth parallel RoI, defined only on the kth mask. Unlike $L_{mask}$, the loss of $L_{cls}$ classification is obtained according to the softmax function, there is category competition between different categories, and $L_{mask}$ is obtained through the corresponding dimension sigmoid function, and a mask is generated for each category, so there is no competition between them between type.

Relying on a specialized classification branch to predict the class label of the output mask, in prediction, sigmoid is not used directly for analysis. First, we select its dimension through the category of the bounding box, then combine the result of this dimension with the sigmoid function, and finally determine whether the result is the mask of this category. According to the prediction result of the sigmoid function of this dimension, it is judged whether the result is the mask of this category.
(1)$$L_{mask}=loss_{1}(x,\mathrm{class})=-x[\mathrm{class}]+\log\left(\sum_{j=k}\exp(x[j])\right)$$

Among them, *x* represents the probability of outputting a multi-classification problem, class represents the index value [0, 1, 2] of the real result, *j* represents the number of classifications, and *k* represents the dimension where the kth mask is located. After the loss function calculation is completed, backpropagation begins. The backpropagation process is essentially a parameter optimization process. For classification tasks, the optimization objects are the weights and biases in the network. For the regression task, the optimization object is the four parameters of *x*, *y*, *w*, and *h* corresponding to the bounding box.
Classification parameters:
(2)$$\begin{array}{c} dw_{2}=\;\mathrm{reg}\times w_{2}+dw_{2}\\ dw_{1}=\;\mathrm{reg}\times w_{1}+dw_{1}\\ w_{2}=-\;\mathrm{epsilon}\times dw_{2}+w_{2}\\ b_{2}=-\;\mathrm{epsilon}\times db_{2}+b_{2}\\ w_{1}=-\;\mathrm{epsilon}\times dw_{1}+w_{1}\\ b_{1}=-\;\mathrm{epsilon}\times db_{1}+b_{1} \end{array}$$
where epsilon represents the learning rate, $w_{1}$ represents the weight from the input to the hidden layer, $w_{2}$ represents the weight from the hidden layer to the output, $b_{1}$ represents the deviation from the input to the hidden layer, $b_{2}$ represents the deviation from the hidden layer to the output, reg is the regularization penalty coefficient value.

Cross entropy loss function:


(3)
$$L_{cls}=loss_{2}(x,\mathrm{class})=-\log\left(\frac{\exp(x[\mathrm{class}])}{\sum_{j}\exp(x[j])}\right)=-x[\mathrm{class}]+\log\left(\sum_{j}\exp(x[j])\right)$$


Among them, *x* represents the probability of outputting a multi-classification problem, class represents the index value of the real result [0, 1, 2], *j* represents the number of classifications, this article is a three-class fault detection, so *j* is 3.

Regression parameters:


(4)
$$\begin{array}{c} t_{x}=d_{x}=\left({\hat{G}}_{x}-P_{x}\right)/P_{w}\\ t_{y}=d_{y}=\left({\hat{G}}_{y}-P_{y}\right)/P_{h}\\ t_{w}=d_{w}=\log\left({\hat{G}}_{w}/P_{w}\right)\\ t_{h}=d_{h}=\log\left({\hat{G}}_{h}/P_{h}\right) \end{array}$$


The regression parameter $P_{i=w,x,y,w,h}$ represents the predicted value. $d_{i=w,x,y,w,h}$ represents the gradient of change. $t_{i=w,x,y,w,h}$ represents the offset calculated according to the target, iGΛ=x,y,w,h represents the change, and each regression parameter update will generate a new ground truth.

SmoothL1 Loss error function:


(5)
$$L_{box}=\frac{1}{N}\sum_{n=1}^{N}t_{n}^{*}\sum_{j\in \{x,y,w,h\}}loss_{3}\left(V_{nj}^{*},V_{nj}\right)$$



(6)
$$loss_{3}\left(V_{nj}^{*},V_{nj}\right)=smooth_{L1}\left(V_{nj}^{*},V_{nj}\right)=\left\{ \begin{array}{l} 0.5{\left(V_{nj}^{*}-V_{nj}\right)}^{2},\left|V_{nj}^{*}-V_{nj}\right|<1\\ \left|V_{nj}^{*}-V_{nj}\right|-0.5,\left|V_{nj}^{*}-V_{nj}\right|\geq 1 \end{array}\right.$$


Among them, N represents the number of anchors, and $t_{n}^{*}$ is the regression of the target frame (1 for the target area and 0 for the background area). $V_{nj}^{*}$ represents the predicted offset. $V_{nj}$ represents ground truth information.

## 3. ARG-Mask RCNN Algorithm

The fault location usually only occupies a small part of the area. To improve the vigilance of the network for small targets, this paper improves the ResNet101 backbone structure and introduces an attention mechanism to focus the model on fault features. The obtained insulators have different degrees of inclination. To generate more personal candidate frames, this paper innovatively proposes a rotation mechanism, which breaks the traditional thinking of generating horizontal anchor frames and overcomes the insufficient target positioning of existing target-detection algorithms. In an accurate bottleneck, at that time, a candidate frame with a rotation angle is generated, which can accurately locate the fault location. In addition, this paper also cleverly introduces the genetic algorithm, which replaces the network parameters originally generated randomly by the genetic algorithm to promote global exploration and improve the accuracy of the model. This section first expounds on the overall framework, then elaborates on the three innovations, and finally points out how these three innovations are applied to the ARG-Mask RCNN algorithm proposed in this paper.

### 3.1. ARG-Mask RCNN Overall Model Framework

The ARG-Mask RCNN network structure consists of four modules as shown in Figure 3, genetic algorithm Figure 3a, the feature extraction Figure 3c, classification and regression Figure 3b, and mask prediction Figure 3d.

Specifically, the first is the genetic algorithm module, which is used to obtain the initial parameter weights and biases required for CNN feature extraction. The second is the feature map module, which is used to extract the target feature map. The original image is extracted through the CNN layer to extract image features, and the RPN layer generates multiple regions of interest. RRoI Align (rotated RoI Align) is a simulation of the RoI Align in Mask RCNN. The rotation mechanism is added. The principle is the same as RoI Align. RRoI Align adds the center rotation parameter, rotates the horizontal candidate frame by a certain angle, adjusts the rotated candidate frame, and finally generates a candidate frame that matches the ground truth. Next is the fully connected layer, which includes two modules: classification and regression, which are used to obtain insulator fault type and location information. The last one is the MPN module to generate mask branches for pixel-level segmentation of insulator fault locations.

### 3.2. ARG-Mask RCNN Backbone Network

The fault location of insulators often occupies a small area in the captured data set. To improve the recognition speed of such small targets, this paper introduces an attention mechanism and improves the ResNet101 structure. Drawing on the idea of transfer learning, for the Conv1 layer as shown in Figure 4, the first layer of ResNet uses a 7 × 7 large convolution kernel to obtain the initial image features in a bigger format. To reduce the calculation amount of the network and improve the efficiency, it is proposed to replace the 7 × 7 convolution kernel of the first layer of ResNet with three layers of 3 × 3 convolution kernels. At the same time, we choose to insert a 7 × 7 Attention between Pre-conv and ResNet101 to improve the recognition speed of the fault location by the module. 

The ResNet101 residual network [28] structure was proposed by He et al. in 2016. The author proposed to construct a deep network through the method of identity mapping. The deep network is copied from the trained shallow layer, and the identity mapping shortcut key. The connection does not add additional parameters and computational complexity, and the network is still trained end-to-end through gradient descent and backpropagation. The actual shooting data set has a large observation area and a large amount of irrelevant information. For example, backgrounds such as tower poles, busbars, trees, etc.; these backgrounds are large and independent, while the area occupied by faults is small and concentrated.

In response to this phenomenon, this paper introduces an attention mechanism, which is very similar to human visual attention, and also enables the machine to select the information that is more critical to the current task goal from a large amount of information. As shown in Figure 5, DANet [29] is chosen to help the model to better select target regions. The net module is a general-purpose lightweight module commonly known as plug-and-play. This module is conducive to improving the accurate screening of insulator minor faults and can obtain more key information.

### 3.3. ARG-Mask RCNN Loss Function

For the photographed infrared insulator map, there are various attitudes, both horizontal and inclined. When a target in the horizontal direction generates a candidate frame, a personal rectangular frame can be generated, and the bounding box generated for an inclined target is much larger than that in the horizontal direction, which means that in the subsequent classification and regression operations, the amount of computation will be greatly increased. To detect these objects with rotation directions in aerial photography, this paper creatively introduces the rotation mechanism into the production of candidate boxes. The difference between rotating target detection and horizontal target detection is that the direction of the target needs to be detected. The predicted result includes the category, position coordinates, length and width, and angle. The Rotated Region of Interest Align (RRoI Align) is based on the Mask RCNN-detection algorithm, adding a rotated Rol extraction module (Rotated Rol), which is divided into two stages. In the first stage, Mask RCNN predicts a rough rotation frame through RPN and horizontal RoI and uses the horizontal RoI feature to predict (x, y, w, h, θ), which represents a rotation angle. The second stage is to extract the features of Rol from the rotation frame of the first stage, and then perform accurate ($x^{\prime }$,$\;y^{\prime }$,$\;w^{\prime }$,$\;h^{\prime }$,$\;\theta^{\prime }$) correction. Rotation Rol feature extraction is implemented based on RoI Align, that is, based on horizontal RoI Align, and each sampling point (x,y) is coordinate offset according to angle θ to obtain (x,y). The final feature extraction of rotation is shown in Figure 6.

In order to achieve the effective separation of target and background, the above Equation (4) will be improved. The arrangement of insulator string facilities is relatively dense, and the acquired data has a large overlap and pick-and-roll situation. It is difficult to achieve accurate instance segmentation for inclined fault locations. Therefore, this paper improves the loss function while improving the backbone network. This paper proposes a rotating anchor frame, which can maintain high localization accuracy and speed for small, inclined objects. That is, a new parameter is introduced into the bounding box loss function to represent the angle of the bounding box on the Y-axis relative to the X-axis, in the range [0, 2], obtained from Equation (4). The improved bounding box is defined as follows.
(7)$$\begin{array}{l} \begin{array}{c} t_{x}=d_{x}=\left({\hat{G}}_{x}-P_{x}\right)/P_{w}\\ t_{y}=d_{y}=\left({\hat{G}}_{y}-P_{y}\right)/P_{h}\\ t_{w}=d_{w}=\log\left({\hat{G}}_{w}/P_{w}\right)\\ t_{h}=d_{h}=\log\left({\hat{G}}_{h}/P_{h}\right) \end{array}\\ t_{\theta }=d_{\theta }={\hat{G}}_{\theta }-P_{\theta } \end{array}$$

$P_{i=w,x,y,w,h,\theta }$ means proposal. $d_{i=w,x,y,w,h,\theta }$ represents the gradient of change. $t_{i=w,x,y,w,h,\theta }$ corresponds to the target as the required offset, iGΛ=x,y,w,h,θ represents the change, and each regression parameter update will generate a new Ground truth.

RRoI Align and RoI Align are essentially the same, except that RRoI Align will have an offset angle for the sampling points during bilinear interpolation. The offset is calculated as:(8)$$\begin{array}{c} x=Sampling_{y}\sin\theta +Sampling_{x}\cos\theta +Center_{w}\\ y=Sampling_{y}\cos\theta +Sampling_{x}\sin\theta +Center_{h} \end{array}$$

Among them, $Center_{w}$, $Center_{h}$ represents the (*x*, *y*) coordinates of the center point, respectively. $Sampling_{x}\;$, $Sampling_{y}$ represents the (*x*, *y*) coordinates of the feature map where the sampling point is located.

The improved loss function: obtained by Equations (1), (3) and (5).
(9)$$L=\frac{\lambda_{1}}{N}\sum_{n=1}^{N}t_{n}^{*}\sum_{j\in \{x,y,w,h,\theta \}}L_{reg}\left(V_{nj}^{*},V_{nj}\right)+\frac{\lambda_{2}}{N}\sum_{n=1}^{N}L_{cls}\left(x,\mathrm{class}\right)+\frac{\lambda_{3}}{N}\sum_{n=1}^{N}L_{mask}\left(x,\mathrm{class}\right)$$

Among them, *N* represents the number of anchors, and $t_{n}^{*}$ is the regression of the target frame (1 for the target area and 0 for the background area). $V_{nj}^{*}$ represents the predicted offset. $V_{nj}$ represents the GT information, *x* represents the probability of outputting a multi-classification problem, class represents the index value [0, 1, 2] of the real result, and $\lambda_{1}\;\lambda_{2}\;\lambda_{3}$ is three hyperparameters that control the balance of the two losses.

### 3.4. ARG-Mask RCNN Parameter Update

The classic convolutional neural network adopts the steepest descent algorithm as the optimizer, and its optimal performance is greatly affected by the initial weight settings of the convolutional layer and the fully connected layer. The genetic algorithm is used to generate multiple groups of initial weights, and the optimal weights are obtained through selection, crossover, and mutation operations. These weights are used as the initial weights of the neural network, and their performance is better than the initial weights randomly selected by the steepest descent algorithm. Considering that the genetic algorithm has the efficient searchability of the global and local optimal solutions, this paper proposes a convolutional neural network combined with a genetic algorithm to optimize parameters to be as close to the global optimum as possible.

The genetic algorithm [30] is used to determine the initial weight of the neural network classifier, as well as the initial position of the bounding box of the regressor and the size of the target box. The weight of the convolution layer in the neural network and the parameters of the bounding box are used as the population individuals of the genetic algorithm, and all combinations of weights and parameters are binary-coded to generate the chromosomes of the genetic algorithm. Then, we perform reselection, crossover, and mutation operations on each chromosome in the population to approach the one with the better weight. To solve the chromosome fitness value, decode the chromosome to obtain a set of initial values, which will be used as the initial value of the neural network and the initial parameters of the generated frame, and the generated initial value will be used to train the neural network classification by using the steepest descent algorithm. The loss function value of the convolutional neural network after training is calculated and used as the fitness value of the corresponding chromosome. To avoid data overfitting, the number of iterations should not be set too large, and the genetic algorithm can be used to mark many local optimal values. For given population size, after performing multiple rounds of the genetic algorithm, the final population can be obtained, which will be used as the initial parameter. The flow chart of the realization of the algorithm is shown in Figure 7.

The training of the neural network is the process of updating the parameters according to backpropagation. At that time, the optimizer will calculate the new value according to the gradient information of backpropagation. Adaptive moment estimation (Adam) and stochastic gradient descent (SGD) are the best deep learning optimizers today, respectively. They have advantages in efficiency and precision. Adam has fast optimization speed in the early stage, while SGD has high optimization accuracy in the later stage. To test the performance of target recognition using the genetic algorithm to generate the initial parameters of the network proposed in this paper, we chose to test the classification task on the CIFAR10 dataset. The experimental results are shown in Figure 8. In the figure, SGD [31] represents the stochastic gradient descent method; GA-GD is the genetic algorithm combined with the gradient descent method proposed in this paper, and the Adam [32] algorithm evolved from SGD. The Adam algorithm has been used in recent years and is widely used in the field of computer vision. The experimental results show that the GA-GD algorithm can quickly complete the classification task. Compared with the other two methods, there are much fewer roundabout processes, and the final recognition accuracy is about 72%, which is better than other methods.

### 3.5. ARG-Mask RCNN Algorithm Implementation Steps

The previous section has described the basic structure of the ARG-Mask RCNN algorithm in detail, including its backbone network, loss function, and parameter update. This section will concatenate these structures, specifically how the loss function is used to optimize the initial parameters, and how these basic structures are stitched together to form the final ARG-Mask RCNN algorithm.

Step 1: Feature extraction according to the filter [33]; the process of layer-by-layer convolution of the original image is completed. As the number of convolution layers increases, image information will also be lost, and the loss of a large amount of information will be extremely unfavorable for the regression task. The backbone network of ARG-Mask RCNN is composed of FPN + ResNet101. FPN effectively retains the basic characteristics of the image by summing and superposing the various convolutional layers of ResNet101. When an image is input, the backbone network first performs noise reduction processing on the original image and then performs scaling and superposition processing on the R, G, and B channels of the image. During feature extraction of the target, the backbone network of ARG-Mask RCNN can calculate the edge information of the target.

Step 2: Calculation of loss function after completing the first step; the processed feature map is sent to the full link layer to complete the classification and regression tasks [34]. In this paper, four-class fault detection is performed on insulators. According to the input feature map, the neural network will predict the probability values of four types of faults. At that time, ARG-Mask RCNN will calculate $L_{cls}$ based on the difference between the predicted result and the real situation. In the regression task, $L_{box}$ is calculated from the difference between the predicted fault location and the true location. In the same way, $L_{mask}$ is calculated.

Step 3: Update of parameters after the calculation of the loss function is completed and back-propagation begins, that is, the process of optimizing the parameters. As a tool to measure the quality of the model’s prediction, the loss function can reflect the gap between the predicted value and the actual value. In Section 3.3, the loss function of ARG-Mask RCNN was described, and its expression was used as the objective function as shown in Equation (10). Taking the parameters in the classification and regression tasks as the object of optimization, such as Equation (11), the update of the parameters is completed by the GA-GD algorithm.
(10)$$f(x)=\frac{\lambda_{1}}{N}\sum_{n=1}^{N}t_{n}^{*}\sum_{j\in \{x,y,w,h,\theta \}}L_{reg}\left(V_{nj}^{*},V_{nj}\right)+\frac{\lambda_{2}}{N}\sum_{n=1}^{N}L_{cls}\left(x,\mathrm{class}\right)+\frac{\lambda_{3}}{N}\sum_{n=1}^{N}L_{mask}\left(x,\mathrm{class}\right)$$
(11)$$\left[\begin{array}{ccccccccc} W_{11} & W_{21} & b_{11} & b_{21} & {\hat{G}}_{x1} & {\hat{G}}_{y1} & {\hat{G}}_{w1} & {\hat{G}}_{h1} & {\hat{G}}_{\theta 1}\\ \vdots & \vdots & \vdots & \vdots & \vdots & \vdots & \vdots & \vdots & \vdots \\ W_{1i} & W_{2i} & b_{1i} & b_{2i} & {\hat{G}}_{xi} & {\hat{G}}_{yi} & {\hat{G}}_{wi} & {\hat{G}}_{hi} & {\hat{G}}_{\theta i}\\ \vdots & \vdots & \vdots & \vdots & \vdots & \vdots & \vdots & \vdots & \vdots \\ W_{1n} & W_{2n} & b_{1n} & b_{2n} & {\hat{G}}_{xn} & {\hat{G}}_{yn} & {\hat{G}}_{wn} & {\hat{G}}_{hn} & {\hat{G}}_{\theta n} \end{array}\right]$$

Among them, $W_{1i}b_{1i}W_{2i}b_{2i}$ is the classification optimization parameter, $W_{1i}b_{1i}$ is the weight and deviation from the input layer to the hidden layer, $W_{2i}b_{2i}$ is the weight and deviation from the hidden layer to the output, $G_{xi}G_{yi}G_{wi}G_{hi}G_{\theta i}$ is the regression parameter, $G_{xi}G_{yi}$ corresponds to the coordinate information of the center of the rectangular frame, and $G_{wi}G_{hi}$ corresponds to the width of the rectangular frame and height, $G_{\theta i}$ corresponds to the rotation angle of the rectangular box.

## 4. Simulation Experiment

In order to test the recognition performance of the ARG-Mask RCNN method proposed in this paper for infrared fault insulators, a comparison experiment with the classical convolutional neural network algorithm is proposed to verify whether the recognition accuracy and speed can be improved. This section will elaborate on the experimental environment, experimental results, and performance tests. The specific experiments are as follows: (1) The ARG-Mask RCNN algorithm is used for edge extraction to separate the insulator from the background. (2) Analyze the fault-detection performance of the ARG-Mask RCNN algorithm, and the test data set contains different fault types. (3) It is proposed to compare the ARG-Mask RCNN algorithm with the classic target-recognition algorithm to verify whether the algorithm proposed in this paper can achieve good performance.

### 4.1. Experimental Environment

The infrared insulator images used in this paper are provided by a China Southern Power Grid Company (Nanning, China), from which 6000 images are selected as the training data set, and the remaining 1000 images are used as the test data set. Each insulator string image contains at least four insulator sheets. In this paper, Labelme labeling software (Labelme v5.0.1) is used to label the insulator fault location and type for training analysis. The software is an image annotation tool developed by the Massachusetts Institute of Technology (MIT) in the United States. Labelme software will generate the corresponding JSON file. The experimental environment used in this paper is shown in Table 1.

The data set required by the network is trained by converting the labeled data set into COCO [35]. The modified ResNet50/101 + FPN model is used in the ARG-Mask RCNN model as the backbone network, and the hyperparameters of the model are obtained by a genetic algorithm. The initial hyperparameters are shown in Table 2.

### 4.2. Experimental Results and Analysis

This section will elaborate on the process of image processing, and show the results of edge extraction, object recognition, and fault detection. Among them, fault detection will be the key content, showing the detection effects of four kinds of faults, and at the same time, it will locate different fault locations.

The quality of the image will determine whether the type of insulator fault can be accurately diagnosed, and the image is a complex outdoor environment affected by noise, which leads to the degradation of the image quality. To improve the detection accuracy of the model, based on image processing technology, a Gaussian filter is used to process the noise of the incoming image [36], and the gradient method, non-maximum suppression, and double threshold are used to extract the image edge [37]. The gradient can obtain the changes of pixels in the region, use non-maximum suppression to retain the nine boundary contours with the largest changes in adjacent pixels, and finally use double thresholds to obtain strong edges greater than the upper limit while retaining candidates between the upper and lower limits. Weak edges, as shown in Figure 9, are the effects of image processing.

Figure 9 shows a total of four original infrared images representing self-explosion faults, low-value faults, zero-value faults, and contamination faults. The first is to extract the edge contour information of the insulator by the edge detection of Marginal check, to provide the basis for the subsequent target detection and fault location. Target detection Background Separation shows the separation of the insulator from the background, treating the pixel as a mixture of multiple Gaussian models, and then assigning a Gaussian model to one class, selecting the insulator model to filter out the background [38]. Target extraction performs pixel segmentation on insulators to obtain detection targets. Abort situation locates the fault location of the insulator to supply grid maintenance personnel for subsequent insulator maintenance.

In this paper, the fault detection of infrared insulators based on the ARG-Mask RCNN method is used, and the faulty insulators are segmented by using the Mask. In Figure 10, red, purple, fluorescent green, and blue represent self-explosion faults, low-value faults, and zero-value faults, respectively—four types of contamination faults.

To better illustrate the effect of the method proposed in this paper on insulator-fault detection, Figure 11 shows the detection effect of four different fault locations. The detected fault rectangle in Figure 11a is inclined, which is good as it is proved that the rotation mechanism proposed in this paper can more accurately detect the position of the faulty insulator sheet, and directly generate a horizontal frame for the horizontally placed target [39].

By analyzing the results in Figure 11, it is found that there are four types of faults: self-explosion fault, low-value fault, zero-value fault, and contamination fault. After changing the fault location in various ways, the ARG-Mask RCNN network can still locate it accurately. The highest recognition rate of self-explosion faults is 96%, the highest recognition rate of low-value faults is 95%, the highest recognition rate of zero-value faults is 96%, and the highest recognition rate of pollution faults is 99%. It can be concluded that the method proposed in this paper can identify a variety of fault types, which greatly consolidates the safe and stable operation of the power grid.

### 4.3. ARG-Mask RCNN Performance Test

To fully demonstrate the insulator fault-detection performance proposed in this paper, Cascade RCNN, SSD, Retina Net, Mask RCNN, and YOLOv3 tiny are used as the control group. Among them, Cascade RCNN is a two-stage target-detection algorithm. It proposes a step-by-step method to integrate the IOU, which effectively solves the problem of low training accuracy with a low threshold and a lack of positive samples with a high threshold. SSD belongs to a single-stage target-detection algorithm. It proposes a method for end-to-end direct target detection. After a single detection, the category and position information of the target can be obtained, which reduces the region proposal stage, so the detection speed is faster. After acquiring the feature map, RetinaNet added the FPN feature pyramid for feature fusion and used focal loss to adjust the loss weight to solve the problem of positive and negative sample imbalance. Mask RCNN is a segmentation method based on pixel suggestion. It obtains image feature maps in an end-to-end manner, which can realize convolution sharing, and then perform classification and regression operations on the feature maps. Due to the addition of a mask prediction network, pixel-level segmentation can be performed. YOLOv3 belongs to the one-stage algorithm, which uses only one CNN to directly predict the categories and positions of different targets [40]. It has obvious advantages in speed. Finally, the proposed method is based on ARG-Mask RCNN. Hyperparameters such as epoch, learning_rate, batch_size, etc., are kept the same in all experimental groups. Four factors, *TP*, *TN*, *FP*, and *FN* [41], and four indicators of precision, recall, accuracy, and technique for order preference by similarity to an ideal solution (TOPSIS) [42] are set up. Definitions of these indicators are provided by (12)–(15).
(12)$$Accuracy=\frac{TP}{TP+TN+FP+FN}$$
(13)$$Precision=\frac{TP}{TP+FP}$$
(14)$$Recall=\frac{TP}{TP+FN}$$
(15)$$TOPSIS=\frac{D_{i}^{-}}{D_{i}^{+}+D_{i}^{-}}$$

Among them, *TP* means that the test result is consistent with the actual result, both of which are the same fault type; *FP* means that the test result is a certain fault insulator and the real result is a normal insulator; *FN* means that the real result is a kind of fault insulator, and the test result shows a normal insulator or a fault category that is inconsistent with the actual result is detected; *TN* means that the actual result and the test result are normal insulators; $D_{i}^{+}$ represents the distance between the various indicators of an object and the maximum value; and $D_{i}^{-}$ represents the distance between various metrics and the minimum value of an object.

It should be noted that TOPSIS is a comprehensive indicator that combines two parameters—accuracy and frames per second (FPS) [43]. The calculation process of the TOPSIS indicator is as follows:(16)$$x_{i}^{*}=\frac{x_{i}-x_{\min}}{x_{\max}-x_{\min}}=\frac{x_{i}-x_{\min}}{x_{\max}-x_{i}+x_{i}-x_{\min}}$$
(17)$$x_{i}^{\prime }=x_{\max}-x_{i}$$
(18)$$Y_{nm}=\left[\begin{array}{cccc} Y_{11} & Y_{12} & \cdots & Y_{1m}\\ Y_{21} & Y_{22} & \cdots & Y_{2m}\\ \vdots & \vdots & \ddots & \vdots \\ Y_{n1} & Y_{n2} & \cdots & Y_{nm} \end{array}\right]$$
(19)$$Z_{ij}=\frac{Y_{ij}}{\sqrt{\sum_{i=1}^{n}Y_{ij}{}^{2}}}$$
(20)$$\begin{array}{cc} Z^{+} & =\left(Z_{1}^{+},Z_{2}^{+},\cdots ,Z_{m}^{+}\right)\\ & =\left(\max\left\{z_{11},z_{21},\cdots ,z_{n1}\right\},\max\left\{z_{12},z_{22},\cdots ,z_{n2}\right\},\cdots ,\max\left\{z_{1m},z_{2m},\cdots ,z_{nm}\right\}\right) \end{array}$$
(21)$$\begin{array}{cc} Z^{-} & =\left(Z_{1}^{-},Z_{2}^{-},\cdots ,Z_{m}^{-}\right)\\ & =\left(\min\left\{z_{11},z_{21},\cdots ,z_{n1}\right\},\min\left\{z_{12},z_{22},\cdots ,z_{n2}\right\},\cdots ,\min\left\{z_{1m},z_{2m},\cdots ,z_{nm}\right\}\right) \end{array}$$
(22)$$D_{i}^{+}=\sqrt{\sum_{j=1}^{m}\omega_{j}{\left(Z_{j}^{+}-z_{ij}\right)}^{2}}$$
(23)$$D_{i}^{-}=\sqrt{\sum_{j=1}^{m}\omega_{j}{\left(Z_{j}^{-}-z_{ij}\right)}^{2}}$$
(24)$$TOPSIS=S_{i}=\frac{D_{i}^{-}}{D_{i}^{+}+D_{i}^{-}}$$

Among them, $x_{i}$ represents each evaluation object, $x_{\max},x_{\min}$ represents the largest and smallest evaluation object in a certain evaluation index, $x_{i}^{\prime }$ represents the forwarded data, $Y_{nm}$ represents the permutation and combination of the forwarded data, and *n* represents the object (the text refers to different methods), *m* represents the evaluation index (the text refers to accuracy and time), $Z^{+},Z^{-}$ represents the maximum and minimum values in each column, and $\omega_{j}$ represents the weight of different indicators (this paper sets the accuracy weight as 0.8, and the time weight as 0.2). $D_{i}^{+}$ represents the distance between each indicator of an object and the maximum value, $D_{i}^{-}$ represents the distance between each indicator of an object and the minimum value, and $S_{i}$ represents the final score.

According to the results of the demonstration, for the extraction of insulators, four parameters of six different detection methods, TP, TN, FP, and FN, are counted. The two indicators of precision and recall are obtained from Equations (13) and (14), respectively, as shown in Table 3.

There are six different detection methods in Table 3: Cascade RCNN, SSD, Retina Net, Mask RCNN, Yolov3 tiny, ARG-Mask RCNN. The precision indicators are 0.868, 0.846, 0.876, 0.961, 0.842, 0.984, and the recall indicators are 0.856, 0.822, 0.894, 0.908, 0.839, 0.988. The experimental results show that the method proposed in this paper is significantly better than other methods for the two indicators of precision and recall.

To further analyze the effect of each method on identifying different fault types [44], the accuracy of fault detection for each method is obtained from Equation (12). Table 4 lists four typical fault types: self-explosion, low value, zero value, and pollution. The detection accuracy was evaluated according to the Equation (15) TOPSIS method. Six different detection methods were taken as the object, and mean accuracy and FPS were used as two indicators for performance analysis.

According to the data in Table 4, the average accuracy of single-stage object-detection methods such as SSD and YOLOv3 tiny is about 72%, which is generally low. The recognition accuracy of Mask RCNN, Cascade RCNN, and the method proposed in this paper is generally higher than that of the single-stage object-detection method, but the image running time is longer. The method proposed in this paper has made a good balance between the recognition accuracy and the processing time of each image [45]. At the recognition accuracy of 97.28%, the processing time of each image is only 0.174 s.

Figure 12 shows the change curve of precision and recall rate during the training process. Figure 12 represents six kinds of Cascade RCNN, SSD, Retina Net, Mask RCNN, YOLOv3 tiny, ARG-Mask RCNN, from left to right and from top to bottom. The abscissa represents the number of training rounds and the ordinate represents the percentage [46]. The recall rates of Cascade RCNN, SSD, Retina Net, Mask RCNN, and YOLOv3 tiny are generally between 60% and 65% at the beginning of training, while the method in this paper performs well, at about 75% at the beginning. In terms of recognition accuracy, the method in this paper is also 70% in the initial stage, while most of the other methods are between 55% and 68%. When the detection performance reaches a plateau, the recognition accuracy and recall of the method in this paper hover within 0.5%, while other methods fluctuate greatly in the steady-state. In comparison, the system of this method is more stable [47]. To sum up, the method in this paper is superior to other methods in terms of accuracy, recall, and stability.

Figure 13 shows the effect of six different methods on fault detection. On the one hand, we can acquire the distribution of the accuracy of fault identification by different methods, and on the other hand, we can acquire the probability of different faults being detected [48]. These two data will be of great reference value for future work. The four graphs in Figure 13 represent the boxplots of the identification accuracy of self-explosion faults, low-value faults, zero-value faults, and contamination faults, respectively. The abscissa represents the accuracy of different methods. The ordinate represents the accuracy degree of distribution [49].

To more intuitively show the evaluation results of the six different methods for insulator fault detection, Figure 14 shows the histogram of the performance comparison of these six methods under each evaluation index. It shows that the method proposed in this paper has obvious advantages in insulator fault identification [50].

In Figure 14, the horizontal axis shows precision, recall, accuracy, and TOPSI, representing the four different performance evaluation indicators. Each indicator includes six insulator fault-diagnosis methods: Cascade RCNN, SSD, Retina Net, Mask RCNN, YOLOv3 tiny, and ARG-Mask RCNN. The vertical axis represents the scores under different indicators of each method. The four performance indicators of precision, recall, accuracy, and TOPSIS of the insulator infrared fault-diagnosis method proposed in this paper are better than other methods, which are 0.984, 0.988, 0.972, and 0.873, respectively. From the error bar in Figure 14, it is found that the method proposed in this paper has the smallest error of these four indicators, which further shows that the ARG-Mask RCNN method has the best performance in the infrared insulator fault-diagnosis method [51].

## 5. Discussion and Future Work

Aiming at the problems of the existing image-recognition algorithms, such as single category, low recognition rate, and slow speed in insulator fault diagnosis, this paper proposes an image-segmentation method based on ARG-Mask RCNN. This method has achieved good results in infrared insulator fault detection. Good results, many different fault types can be detected, and the location of the faulty insulator string can be precisely identified. The main method is to use the genetic algorithm to obtain the initial hyperparameters required by the network, which solves the problem that it is difficult to obtain the global optimal solution through random selection combined with the steepest descent algorithm. Modify the backbone network model to reduce the time for small target recognition; the rotating target detection algorithm improves the accuracy of fault location. The experimental results show that the method proposed in this paper can effectively solve the problems of the current insulator detection system, such as single function, low accuracy, slow speed, and difficulty in dealing with harsh environments.

Many factors cause the failure of insulators, but most of them are determined by natural factors. What we can do is to find it as soon as possible and reduce unnecessary losses. Deep learning methods are popular in the field of insulator fault identification. Although the method proposed in this paper has achieved good results, there are still some limitations worthy of further study: (1) In the actual fault detection, the influence of various types of bad weather should be considered. For example, in the background of rainy and dense fog, the detection accuracy of the model will drop slightly. (2) There are slight differences between some faults, which will normally cause the network to fail to identify such faults, and even confuse faults with similar characteristics. (3) There are many kinds of faults of insulators. This paper only covers four common fault detections: self-explosion fault, contamination fault, zero fault, and damage fault. For some uncommon types of failures, it is not yet possible to identify them. The future research direction should continue to expand the data set, improve the recognition rate under various complex environmental backgrounds, mine the differences in the characteristics of different fault types, and subdivide the fault types to highlight the problems of confusion of similar categories. Finally, it is hoped that the method in this paper can be helpful for the construction of smart grids in my country.

## Figures and Tables

**Figure 1 sensors-22-04720-f001:**
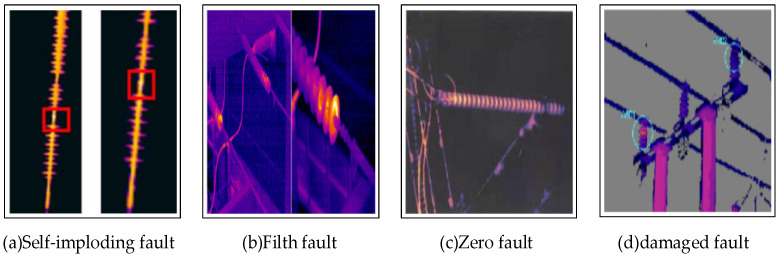
Four typical infrared faults.

**Figure 2 sensors-22-04720-f002:**
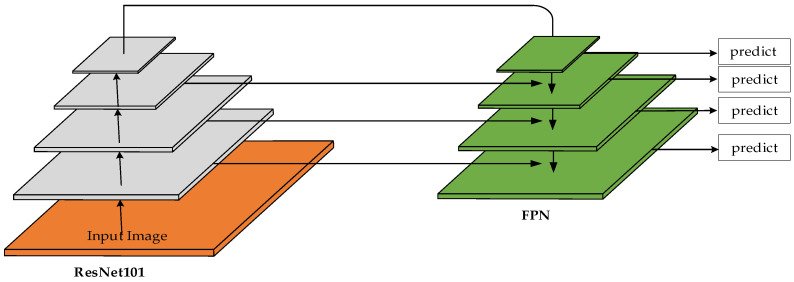
Feature Pyramid Network.

**Figure 3 sensors-22-04720-f003:**
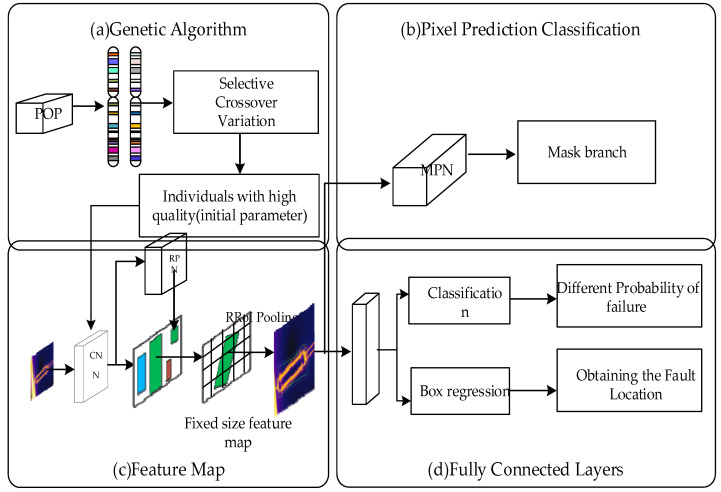
ARG-Mask RCNN framework.

**Figure 4 sensors-22-04720-f004:**
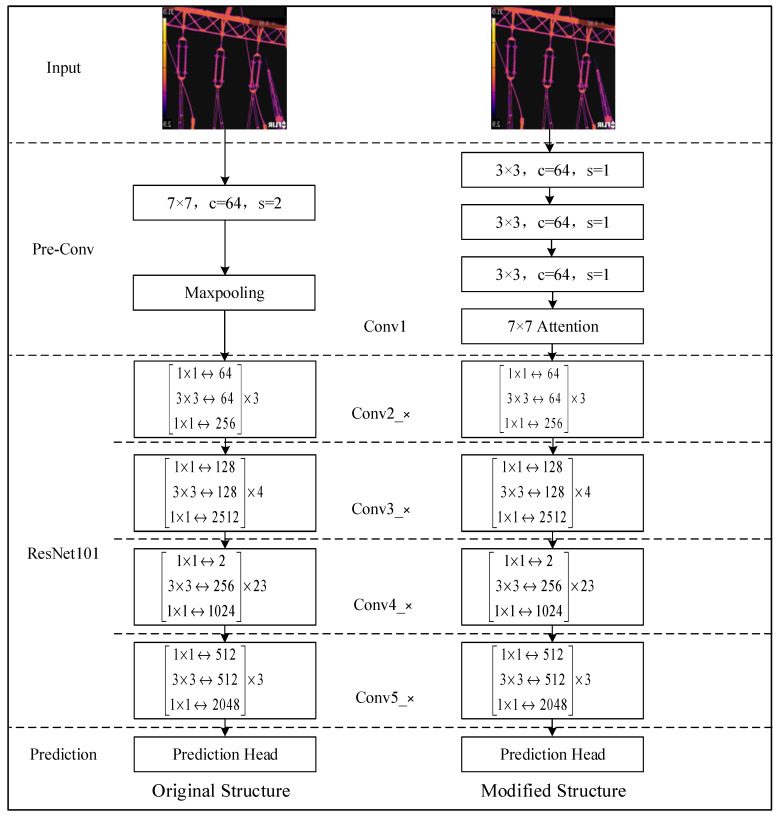
ARG-Mask RCNN backbone network structure.

**Figure 5 sensors-22-04720-f005:**
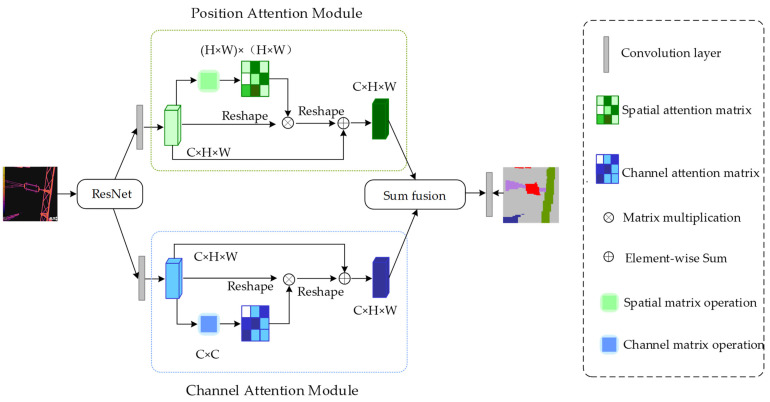
DANet module structure diagram.

**Figure 6 sensors-22-04720-f006:**
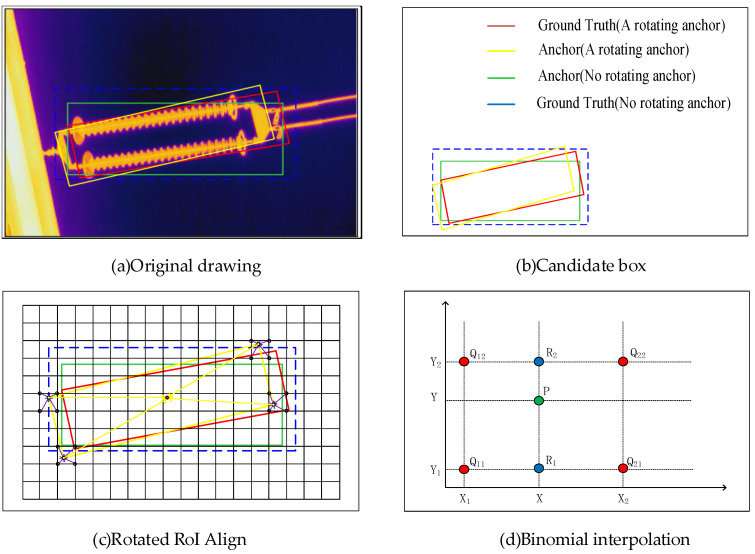
Rotational object refinement extraction module.

**Figure 7 sensors-22-04720-f007:**
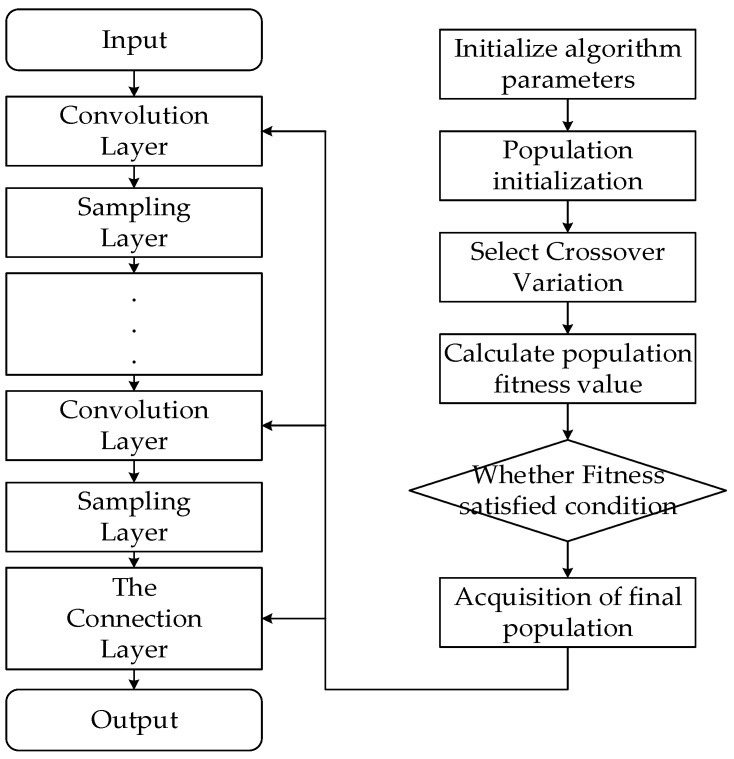
GA-CNN implementation flowchart.

**Figure 8 sensors-22-04720-f008:**
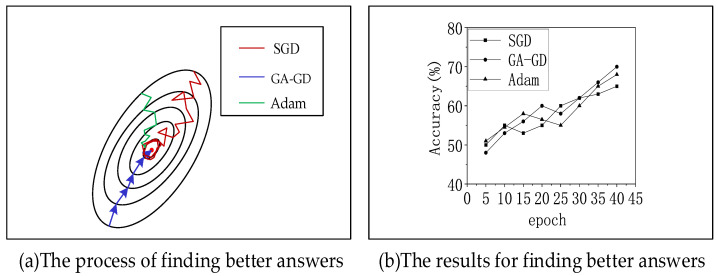
GA-GD test results.

**Figure 9 sensors-22-04720-f009:**
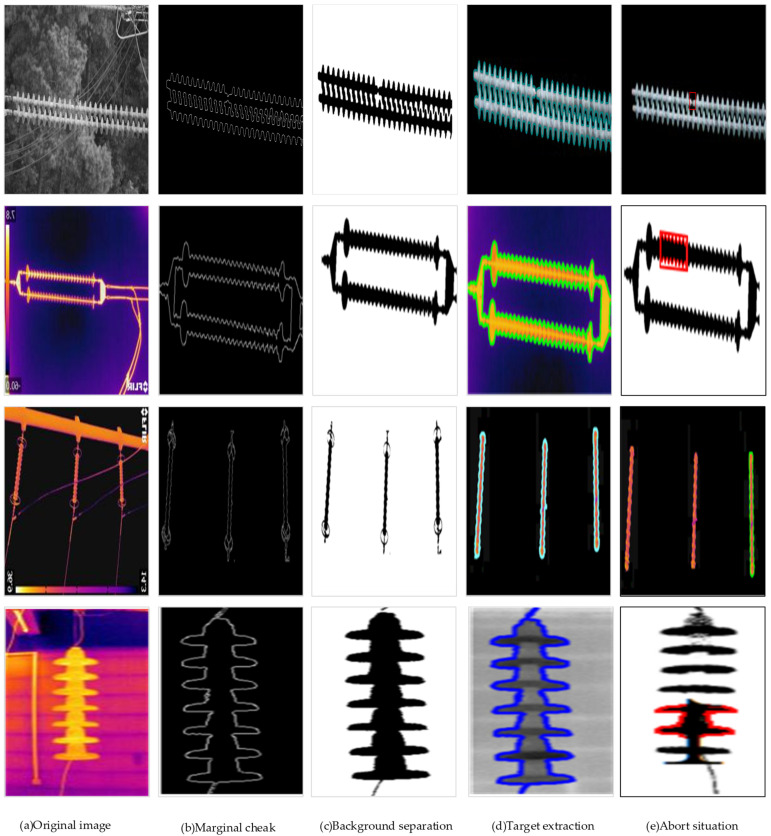
Visualization of the image processing process.

**Figure 10 sensors-22-04720-f010:**
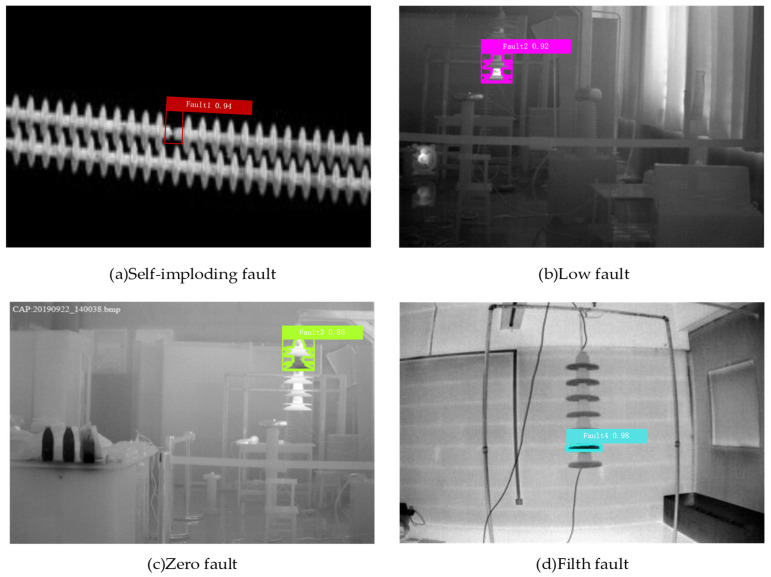
Four typical troubleshooting results.

**Figure 11 sensors-22-04720-f011:**
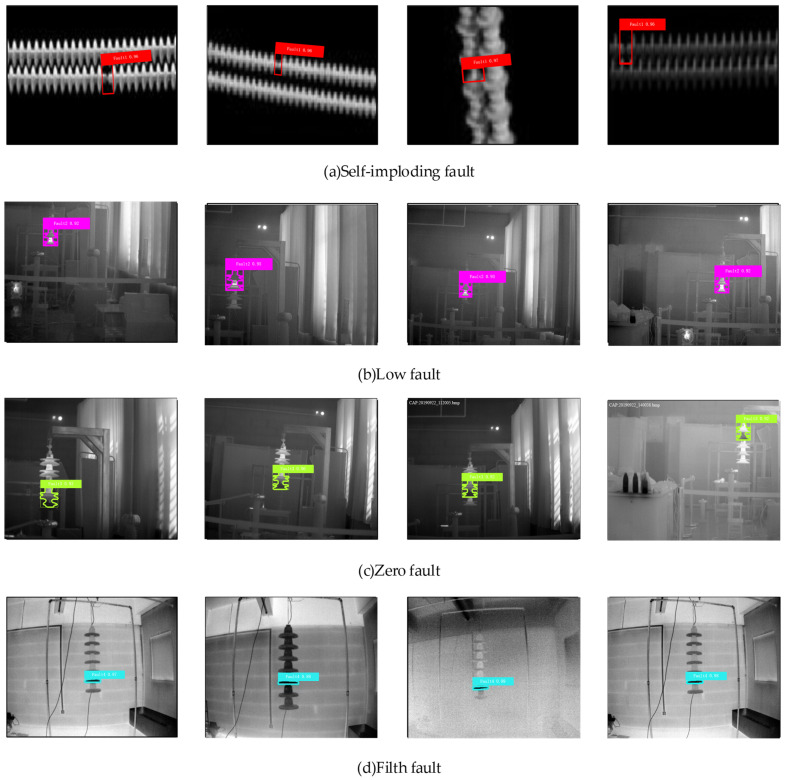
The effect of fault identification at different positions.

**Figure 12 sensors-22-04720-f012:**
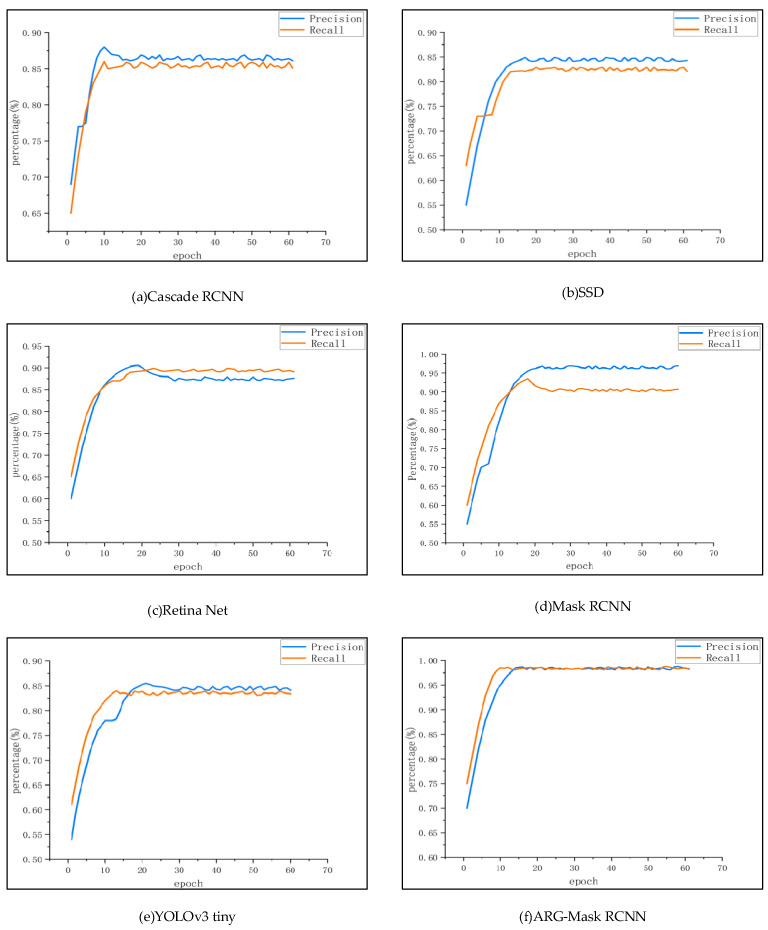
Precision and Recall variation curves for six different methods.

**Figure 13 sensors-22-04720-f013:**
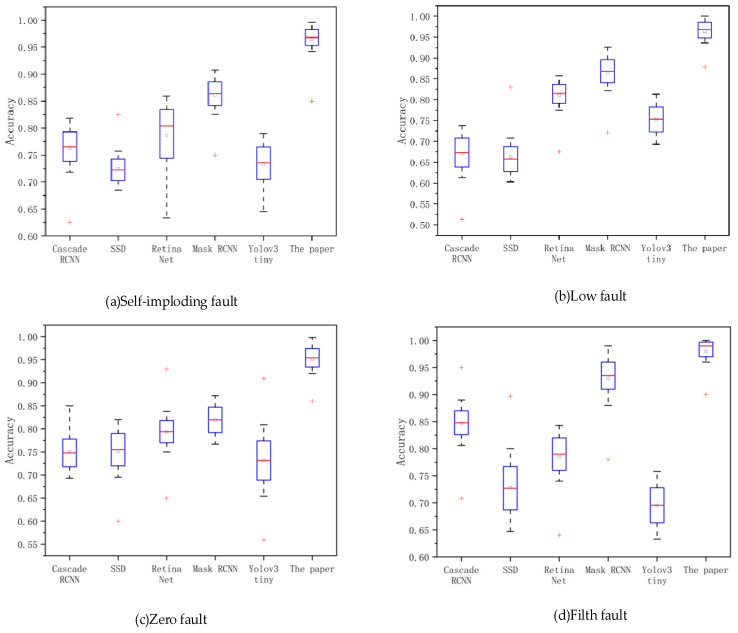
Boxplot of recognition accuracy of different methods. The red “+” represents outliers in the detection results. Each box has two black “−”, one above and the other below, representing the upper and lower limits of the box, respectively. The blue rectangle has upper and lower lines, which represent the upper and lower quartiles, respectively. There is also a short red line in each rectangle, which represents the median in the results. In addition, there is a small blue square in the center of each bin, which represents the mean across the set of results.

**Figure 14 sensors-22-04720-f014:**
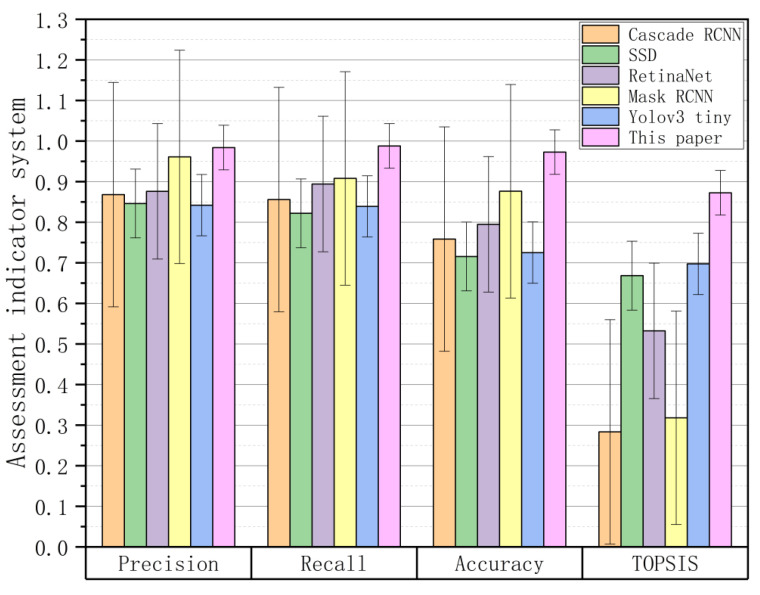
Bar graph comparing the performance of different methods.

**Table 1 sensors-22-04720-t001:** Experimental environment.

Project	Model or Parameter Value
Central Processing Unit (CPU)	Intel i5-7300HQ
RAM/GB	128
Graphics Processing Unit (GPU)	An RTX 3080Ti
Operating System	Window 10
Software Environment	Anaconda3, Cuda11.3, Python3.7
Development Tools	Pycharm
Deep Learning Libraries	PyTorch

**Table 2 sensors-22-04720-t002:** Training parameters.

Parameter	Value
weight decay	0.0001
learning rate	0.001
number of iterations	100
number of training rounds	60

**Table 3 sensors-22-04720-t003:** Comparison of six different detection methods.

Method	Backbone	True Positive (TP)	False Positive (FP)	False Negative (FN)	True Negative (TN)	Precision	Recall
Cascade RCNN	ResNet-101 + FPN	125	19	21	0	0.868	0.856
SSD	VGG-16	231	42	50	0	0.846	0.822
Retina Net	ResNet-101 + FPN	254	36	30	0	0.876	0.894
Mask RCNN	ResNet-101 + FPN	268	11	27	0	0.961	0.908
Yolov3 tiny	DarkNet-53	354	66	68	0	0.842	0.839
ARG-Mask RCNN	Improved ResNet-101 + FPN	316	5	4	0	0.984	0.988

**Table 4 sensors-22-04720-t004:** Six different methods for accuracy and TOPSIS scoring.

Class	Cascade RCNN	SSD	Retina Net	Mask RCNN	YOLOv3 Tiny	ARG-Mask RCNN
Self-imploding fault (%)	76.96	72.63	79.69	87.65	73.42	97.66
Low fault (%)	67.32	64.38	81.47	86.12	74.38	96.82
Zero fault (%)	75.31	75.59	79.46	82.73	73.91	95.46
Filth fault (%)	83.81	73.64	77.34	94.02	68.33	99.18
Mean Accuracy (%)	75.85	71.56	79.49	87.63	72.51	97.28
FPS	1.84	5.97	4.56	3.27	6.41	5.75
Times	0.54	0.17	0.22	0.31	0.16	0.17
TOPSIS	0.2834	0.6684	0.5324	0.2180	0.6973	0.8725

## Data Availability

The data in this study are owned by the research group and will not be transmitted.

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
