# Peer review of "ARG-Mask RCNN: An Infrared Insulator Fault-Detection Network Based on Improved Mask RCNN"

_sensors, 2022, doi:10.3390/s22134720_

Round 1
Reviewer 1 Report
See attached file.

Author Response
Response to Reviewer 1 Comments
We are very grateful to the editors and reviewers for your valuable comments on our manuscript, which has greatly helped to improve the quality of our manuscript. We re-edited various parts of the manuscript based on the reviewer’s comments. Below I will give a point-to-point response to each review.
Main feedback:
Point 1: Why are object detection and segmentation models being used but only evaluated with classification metrics?
Response 1:Thank you very much for your questions about our manuscript. First, we need to clarify that we propose to use Precision, Recall, Accuracy, and TOPSIS as metrics to judge model performance. They were originally developed to solve classification problems, but are now widely used for segmentation tasks as well. In the document "Automatic Fault Diagnosis of Infrared Insulator Images Based on Image Instance Segmentation and Temperature Analysis". In the insulator segmentation task, the author also used Precision and Recall indicators. It is worth noting that TOPSIS is a comprehensive indicator for our evaluation model, which combines two important factors of time and accuracy. Based on the processing time and detection accuracy of each image, TOPSIS will conduct a comprehensive score.
In addition, for the metrics of the segmentation model you mentioned, since YOLOv3 and SSD do not involve the content of the mask, they do not participate in the comparison of segmentation. But all models involve classification tasks. We are not trying to confuse object detection and segmentation models, our focus is being able to detect failures efficiently. Existing models cannot balance real-time performance and accuracy, so we compare who has higher accuracy and real-time performance. Other metrics are irrelevant as long as the fault detection problem is solved. It is worth mentioning that we also compared the segmentation effects of Mask RCNN and ARG-Mask RCNN, and the latter outperformed the former. For the overall logic of the article, it should not appear in the article.
Point 2:Hyperparameters, parameters, and performance metrics are all called “parameters”, which lacks clarity and makes the ideas being expressed difficult to understand.
Response 2:It has to be said that you helped us point out such nuanced issues in the manuscript. It is true that we have not been able to clearly express the meaning of these terms, which is extremely unfavorable for readers to understand the content of the article. To this end, we redefine the positions where these nouns first appeared, and the specific modifications are as follows:
- Parameters are generally used as variables automatically learned by the model, and its update is provided by the model. For example, weights and biases in deep learning.
- Hyperparameters are parameters used to define a model, often set by humans. For example, epochs in deep learning, learning rate, number of iterations, number of neurons in each layer, Batch_size.
- The performance indicators are set by us to judge the performance of the model for insulator fault identification. This paper sets four indicators: Precision, Recall, Accuracy, and TOPSIS.
Point 3:The authors claim to find global optimum solution for hyperparameters (as far as I can tell, based on the issues with terminology) with a genetic algorithm but offer no proof of this.
o There is little indication this is true. The parameters learned by SGD would impact
whether they are optimal, and we cannot know if that result is optimal even if it were.
o Genetic algorithms for hyperparameter search is common and it sometimes helps, but they are heuristic and provide little likelihood of finding global optima.
Response 3:We greatly appreciate your pointing out the issues in the manuscript. First of all, we need to explain that this paper uses the global search ability of the genetic algorithm to optimize our model, so that the results of the model run as close to the global optimal solution as possible. The objects we optimize are hyperparameters in deep learning, such as epochs, learning rate, number of iterations, Batch_size, etc. At present, these parameters are all artificially set, and since there is manual participation, there must be its limitations. Take epochs as an example, if the value is too large, the training time will be affected, and if the value is too small, the accuracy requirements will not be met. If we continuously debug these parameters, this requires re-running the code. However, this process is extremely time consuming. As we all know, the larger the data set, the longer the training time, and it takes half a month for each modification. Therefore, we consider whether we can use the global search ability of the genetic algorithm to determine these hyperparameters.
Specifically, we use epochs, learning rate, number of iterations, the number of neurons in each layer, and Batch_size as the population of the genetic algorithm, and the loss function as the objective function. In order to make the article as a whole logical and clear, we did not elaborate the process of selection, crossover, and mutation. In Ref. [25], the authors describe the genetic algorithm in detail. Furthermore, we also conduct experiments on the CIFAR10 dataset. It is well known that gradient descent is a commonly used optimizer in deep learning. We selected "Stochastic Gradient Descent (SGD)" and "Adaptive moment estimation (AdamSGD)" as controls. As we all know, they are the best optimizers in the field of deep learning, and even the latest YOLOv5 algorithm uses the Adam optimizer. We analyze from two perspectives of finding the optimal path and training results. From Figure 8, we find that in finding the optimal path, both SGD and Adam have a lot of detours, and finally stay at a position far from the optimal path.
The genetic algorithm proposed in this paper combined with the gradient descent method (GA-GD) reduces many detours and approaches the optimal solution as much as possible. From the test results, when the epoch is 50, the GA-GD algorithm has the highest accuracy. It should be emphasized here that the hyperparameters of SGD and Adam are randomly generated. For what you mentioned: "The genetic algorithm optimizes hyperparameters with little chance of finding the global optimum". I have to say that your rigorous and serious attitude towards science is worth learning from. We are indeed a little extreme in our language. We searched for these extreme languages throughout the article, with modifications as follows:
The genetic algorithm makes the results obtained by the model as close to the global optimum as possible.
Point 4:Grammar, punctuation, spacing, and formatting of mathematics are problematic throughout the paper.
Response 4:It's shocking to each of us to see you ask such detailed questions one by one. We really appreciate your asking such detailed questions, which obviously helps improve the quality of the manuscript. At the same time, we apologize for the grammatical errors in the article, such low-level mistakes will not be tolerated. In future work, we must correct these low-level errors. Please accept our most sincere thanks again, and we have revised the entire article as follows:
Abstract
Point 5:Lines 14-19: it’s not grammatically correct
Response 5:Thank you very much for your question, we have changed lines 14-19 as follows:
For the backbone network of Mask RCNN, the structure of Residual network 101(ResNet101) is improved, and the attention mechanism is added, which makes the model more alert to small targets and can quickly identify the location of small targets; Improve the loss function, integrate the rotation mechanism into the loss function formula, generate An anchor frame with a rotation angle is used to accurately locate the fault location; the initial hyperparameters of the network are improved, and the Genetic Algorithm Combined with Gradient Descent (GA-GD) algorithm is used to optimize the model hyperparameters, so that the model training results are as close to the global best as possible.
Point 6:Line 16: can you be more specific about how the loss function is improved?
Response 6:Thank you very much for your question. The improvement of the loss function is a major innovation of this paper, which we will introduce in detail in Section 3.3. The details are as follows:
(a) The previous function
(b)The improved function
For the photographed infrared insulator map, there are various attitudes, both horizontal and inclined. When a target in the horizontal direction generates a candidate frame, a personal rectangular frame can be generated, and the bounding box generated for an inclined target is much larger than that in the horizontal direction, which makes in the subsequent classification and regression operations, The amount of computation will be greatly increased. To detect these objects with rotation directions in aerial photography, this paper creatively introduces the rotation mechanism into the production of candidate boxes. The difference between rotating target detection and horizontal target detection is that the direction of the target needs to be detected. The predicted result includes the category, position coordinates, length and width, and angle. Rotated region of Interest Alogn (RRoI Align) is based on the Mask RCNN detection algorithm, adding a rotated Rol extraction module (Rotated Rol), which is divided into two stages. In the first stage, Mask RCNN predicts a rough rotation frame through RPN and horizontal RoI and uses the horizontal RoI feature to predict (,,,,), which represents a rotation angle. The second stage is to extract the features of Rol from the rotation frame of the first stage, and then perform accurate (,,,,) correction. Rotation Rol feature extraction is implemented based on RoI Align, that is, based on horizontal RoI Align, each sampling point (,) is coordinate offset according to angle to obtain (,) The final feature extraction of rotation is shown in Figure6..
In order to achieve the effective separation of target and background, the above formula (4) will be improved. The arrangement of insulator string facilities is relatively dense, and the acquired data has a large overlap and pick-and-roll situation. It is difficult to achieve accurate instance segmentation for inclined fault locations. Therefore, this paper improves the loss function while improving the backbone network. This paper proposes a rotating anchor frame, which can maintain high localization accuracy and speed for small inclined objects. That is, a new parameter is introduced into the bounding box loss function to represent the angle of the bounding box on the Y-axis relative to the X-axis, in the range [0, 2], obtained from formula (4). The improved bounding box is defined as follows.
means proposal. represents the gradient of change. corresponds to the target as the required offset, represents the change, and each regression parameter update will generate a new Ground truth.
The improved loss function: obtained by formula (1)(3)(5)
Among them, N represents the number of Anchors, and is the regression of the target frame (1 for the target area and 0 for the background area). represents the predicted offset. represents the GT information, x represents the probability of outputting a multi-classification problem, class represents the index value [0, 1, 2] of the real result, and is three hyperparameters that control the balance of the two losses.
Point 7:Line 22:“my” should be “our”
Response 7:Thank you very much for pointing out the problems in the manuscript, we agree with you. The specific modifications are as follows:
and reliable operation of our country's power system.
Introduction
Point 8:Line 45: “the” should not be capitalized
Response 8:Thank you very much for pointing out the problems in the manuscript, we agree with you. The specific modifications are as follows:
The results show that the location of the peeling defect can be accurately determined by detecting the propagation attenuation coefficient of the waveguide.
Point 9: Lines 64-65: inconsistent spacing
Response 9: For example, cascade of the regions with cnn features (Cascade RCNN) [15], single shot multibox detector (SSD) [16], RetinaNet [17], Mask RCNN [18], you only look once(YOLO) [19] and other methods.
Point 10: Line 65: cite MnasNet paper:
Tan, M., Chen, B., Pang, R., Vasudevan, V., Sandler, M., Howard, A., & Le, Q. V. (2019). Mnasnet: Platform-aware neural architecture search for mobile. In Proceedings of the IEEE/CVF Conference on Computer Vision and Pattern Recognition (pp. 2820-2828).
Response 10:Thank you very much for your suggestions. We agree that citing the MnasNet paper makes the logic of the article more rigorous. The specific modifications are as follows:
Liu et al. [20] proposed an improved SSD insulator detection algorithm , using a lightweight network MnasNet [21] as a feature extraction network, and then using a multi-scale fusion method to fuse the feature maps.
- Tan, M.; Chen, B.; Pang, R.; Vasudevan, V.; Sandler, M.; Howard, A.; Le, Q. V. In Mnasnet: Platform-aware neural architecture search for mobile, Proceedings of the IEEE/CVF Conference on Computer Vision and Pattern Recognition, 2019; 2019; pp 2820-2828.
Point 11: Line 67: grammar: “Results It show…”
Response 11:Thank you very much for pointing out the problems in the manuscript, we agree with you. The specific modifications are as follows:
The results show that the algorithm can effectively detect the position of the insulator, and has the advantages of small model size and fast detection speed.
Point 12: Line 77: Cite K-Means++ Arthur, D. & Vassilvitskii, S. (2007). k-means++: the advantages of careful seeding. Proceedings of the eighteenth annual ACM-SIAM symposium on Discrete algorithms. Society for Industrial and Applied Mathematics Philadelphia, PA, USA. pp. 1027–1035.
Response 12:Thank you very much for pointing out the problems in the manuscript, we agree with you. The specific modifications are as follows:
and used the improved K-means++ algorithm [24] to redesign the number and size of anchor boxes.
- Arthur, D.; Vassilvitskii, S. k-means++: The advantages of careful seeding; Stanford: 2006.
Point 13: Line 86: Want et al. must have changed the view of the task from object detection to image segmentation since all previously mentioned methods are object detectors but Mask RCNN is a segmentation algorithm – this seems like a significant change. I think this should be discussed. Without this distinction, it’s not clear why one would use Mask RCNN rather than Faster RCNN.
Response 13:
We all agree with you that all the methods mentioned above are object detectors, but Mask RCNN is a segmentation algorithm, which is really a big change. In this regard, we have made a key explanation between the two algorithms, as follows:
The above algorithm belongs to the object detection algorithm whose output is in the form of a bounding box. For large targets such as insulators, the algorithm can complete the positioning task. However, it is obviously difficult to accomplish multi-type fault identification. The reasons are as follows: as we all know, the insulator failure caused by cracks takes up such a small area. If we continue to use this algorithm to generate anchor frames, we can only determine the approximate location of cracks. However, as a segmentation algorithm, Mask RCNN can accurately The edge positions of the cracks are segmented, and our experimental results can prove this.
In addition, for the Faster RCNN you mentioned, it is obvious that Mask RCNN is an improved version of it, and both algorithms were proposed by He Kaiming. The author modified RoI Pooling to RoI Align based on Faster RCNN. RoI Align uses bilinear interpolation instead of the previous forensic operation, so that each RoI gets a feature map, which can better align the RoI regions on the original image. So we didn't choose Faster RCNN.
Point 14: Lines 108-110: The authors claim:
“After that, it is proposed to improve the initial parameters, and the updated parameters originally generated by Mask RCNN are now generated by a genetic algorithm, to obtain the global optimal solution improve the identification accuracy of faulty insulators.” This is very unlikely to be true. There is no known way to verify this for the non-convex optimization problem the algorithm attempts to optimize.
Response 14:
Thank you very much for your question, I have to admit that our statement here is indeed wrong, and we apologize here. At the same time, we check this syntax for errors throughout. Our original intention was, "Then propose to improve the initial hyperparameters, and now use the genetic algorithm to replace the randomly generated hyperparameters of Mask RCNN, so that the results of the model are as close to the global optimum as possible and the identification accuracy of faulty insulators is improved.". We have made revisions to the latest manuscript. It should be emphasized here that hyperparameters can be set manually. If the initial value of the hyperparameter is not set, it is randomly generated by the model. Parameters are generated by the model without human intervention. In Response 2, we highlighted the difference between parameters and hyperparameters. Meanwhile, it is also stated in the latest manuscript.
Basic Theory
Point 15:
I’m not sure this section is titled appropriately… there’s not much theory being discussed. In all paragraphs in section 2.1, the spacing is inconsistent – random spaces or periods are in different places. The numbers with circles are a little unusual, maybe using roman numerals for the second-level list would be better.
Response 15:We agree with you that the basic theory does not fit. In this regard, we revise the basic theory into related work. For the content of 2.1, we have made detailed modifications, as follows:
The infrared data source will be explained below, firstly indicating the characteristics of infrared imaging technology, then introducing the four types of faults with the highest appearance rate, and finally emphasizing the matters needing attention when collecting infrared data. These works will play a crucial role in the labeling of the dataset and later training.
- Compared with other fault diagnosis data, infrared imaging data has the following outstanding characteristics. â… .The data collection is convenient and the work efficiency is high. It only takes a few hours to complete the collection of a large amount of data with the drone. â…¡.During the actual inspection, it can be obtained without touching the equipment to avoid product damage caused by improper operation during inspection. â…¢.A variety of typical faults can be detected, and the location of the faulty insulator sheet and the degree of damage can be located. â…£.Infrared light can detect the internal characteristics of the equipment when it is running. The location of the fault can be identified by the color of the light, which is related to its fault principle, while it is difficult to find faults caused by cracks and internal defects with visible light.
(2) To detect a variety of different faults of insulators, it is necessary to determine which type of fault is caused when the data set is marked. The quality of the data set will directly affect the identification of faulty insulators. To avoid confusion and the inability to identify different fault types, the following will introduce the characteristics of four typical infrared faults in detail.
The fault classification of inferior insulators is shown in Figure1. â… .The type of fault caused by self-explosion can cause some insulator pieces to be missing. â…¡.Stain and dust fault, common surface stains such as ice and branches will cause the surface temperature of the insulator to exceed 1000 degrees Celsius. â…¢.Zero-value insulator, the surface of the zero-value insulator fault is dark red. â…£.The insulator sheet is broken, and the temperature difference between the phases of the insulator sheet at the fracture is greater than 18 degrees Celsius.
Point 16: Line 163: “segmentation to achieve human pose estimation.”
It’s true the original paper used Mask RCNN for Pose estimation, but that’s just one example – Mask RCNN’s state-of-the-art performance on COCO dataset segmentation at the time of publication is more relevant.
Response 16:We strongly agree with you that Mask RCNN's state-of-the-art performance on COCO dataset segmentation at the time of publication is more relevant. We have also done a lot of tests on the COCO dataset, but here we would like to emphasize the role of Mask RCNN that was originally proposed.
Point 17: Line 206: spacing issue
Response 17:Thank you very much for pointing out the problem in the article, we have removed the spaces.
Point 18: Line 209: loss function should be described as a hyperparameter
Response 18:Thank you very much for pointing out the problem in the article. First, it needs to be emphasized that hyperparameters are set by humans or randomly generated by the model. The loss function expresses the degree of the gap between the prediction and the actual data. It is generated by the result of model training. There is an essential difference between the two. The loss function can only represent one important factor.
Point 19: Line 210: loss function does impact convergence, but it importantly controls the objective of the network.
Response 19:Thank you very much for your point, we have modified line 210 as follows:
The loss function determines the convergence effect of the model to a large extent, and also controls the objective of the network.
Point 20: Line 214: “Unlike L_cls, the loss of L_cls classification is obtained according to…”
Response 20:Thank you very much for pointing out the problem in the article.We have modified L_cls to L_mask.
Point 21: Line 216: sigmoid is misspelled
Response 21:Thank you very much for pointing out the problem in the article.We have modified sigmod to sigmoid.
Point 22: Line 219: grammar at the end, the meaning isn’t clear
Response 22:Thank you very much for pointing out the problem in the article.We have rewritten the sentence as follows:
In prediction, sigmoid is not used directly for analysis. First select its dimension through the category of the bounding box, then combine the result of this dimension with the sigmoid function, and finally determine whether the result is the mask of this category.
Point 23: Line 220: what does it mean to “select the dimension” of the output ask? This doesn’t make muchsense. I think the authors mean it selects the nearest basis vector after feeding the image through thenetwork.
Response 23:The model needs to be able to identify four different failures, each with its own dimension. For example, now we need to find the positions of dogs, cats and people from a picture, which is a three-class problem. Dogs, cats, and people all have their own dimensions, and the sigmoid function makes the final prediction based on the dimension of each label.
Point 24: Equation (1) is not correct…, it’s not even clear what this formula means as it is written now. What does x[class] even mean? The summation has no maximum index.
Response 24:x[class] represents the probability of predicting the occurrence of each fault, since we are working on a four-class problem, x[0] represents the probability that the input is a self-exploding fault, x[1] represents the probability that the input is a low-value fault, x[2 ] means zero value and x[3] means dirty. The model predicts the probability of these four failures for the input image. It should be emphasized that x[0]+x[1]+x[2]+x[3]=1. The class label with the highest probability value is selected as the prediction result. For example, now predict the probability values of all categories, x[0]=0.2, x[1]=0.1, x[2]=0.6, x[3]=0.1. The model finally predicts that the fault type is zero-value fault (x[2]). It is important to emphasize that if the model predicts that the four failures have similar probabilities, then this means that the failure category cannot be identified. If one of the x[class] values is large and the other three are small, the model predicts well. In this way, the expression of the loss function is not difficult to explain.
Point 25: Line 223: “probability of outputting a multi-classification problem” What does this mean?
Response 25:For four different types of faults, the model will predict the probabilities of these four types of faults based on the input pictures, which we have also modified in the paper.
Point 26: Lines 231-235: This section is quite confusing. It’s not clear what layers are being referred to. Apparently there are only two layers, but Mask RCNN has dozens of layers. Beyond that, even if there are two layers in this classification part, there are many weights between the layers, so having just two derivatives doesn’t make sense. Use of “reg” rather than is unusual. “Epsilon” shouldn’t be used for a learning rate as it is generally used for other purposes like tiny numbers—better to use . However, this is presumably a function of time as most training processes these days use an evolving learning rate. Equation (3): what is the input to L_cls?
Response 26:The question asked by the reviewer is about the content of the neural network algorithm. The "NN" in Mask RCNN stands for Neural Network. A neural network generally includes an input layer, a hidden layer and an output layer. Neural network is a complex network system formed by a large number of neurons connected to each other. It reflects the process of model learning. Each time it learns, the weights (w1) and biases (b1) of the input layer are transferred to the hidden layer. At the same time, the weights (w2) and biases (b2) of the hidden layer are transmitted to the output layer. This process is continuous, and the parameters (w1, b1, w2, b2) are constantly updated. In order for the reader to fully understand how the model updates the parameters, Equation 2 gives the updates of the weights (w) and biases (b). It should be noted that there are not only two derivatives here, but also represent a class of problems. We want our readers to be aware of their changes. The purpose of using "reg" is to normalize the data. A large number of experiments have proved that regularization can improve the training speed, speed up the convergence process, prevent the model from overfitting, and alleviate the gradient explosion during training to a certain extent. .In mathematics, epsilon represents a positive number close to 0, but the learning rate in deep learning tends to be a small positive number. epsilon can express the meaning of learning rate very well. However, the learning rate cannot be a function of time. It is set by humans. After each iteration, the error function of the model is used to judge. If the error rate decreases, the learning rate can be increased. Usually a small value is chosen as the learning rate at the beginning.
The input of the L_cls function is the probability value, and this paper is a four-class fault detection. The model predicts the probability of these four failures, and determines the label with the highest probability as the final result.
Point 27: Lines 240-244; the authors first say they will predict x, y, w, h but now switch to these alternatives. Which ones are being used? Why is “Ground” capitalized?
Response 27:Our model needs to complete two tasks, one is classification and the other is regression. The model needs to determine the fault category, and also needs to locate the fault location with an anchor frame. Both processes are necessary. "Ground" G has been changed to lowercase
Point 28: Line 245: there are no spaces
Equations (5)-(6): the V’s are not defined, formatting does not show accurate mathematical notation— recommend rewriting in a proper LaTeX equation environment.
Response 28:Thank you very much for your question, we added a space on line 245 and re-edited the formula.
Among them, N represents the number of Anchors, and is the regression to the target frame (1 represents the target area, 0 represents the background area). represents the predicted offset. represents GT information, x represents the probability of outputting a multi-classification problem, class represents the index value [0, 1, 2] of the real result, and is three hyperparameters used to control the balance of the loss.
Point 29: Line 246: Why is “Anchor” capitalized? Beyond that, anchor boxes need to be defined here.ly
Response 29: A is lowercase. Anchor is a series of bounding boxes obtained by image sampling in the segmentation task. It is mainly for regression tasks, that is, the anchor box needs to coincide with the ground truth as much as possible.
ARG-Mask RCNN
Point 30: Line 252: “to enhance the favorability of fault features”
Recommend writing “to focus the model on fault features”
Response 30:Thank you very much for your suggestion, we all agree with you. and revised in the manuscript.
Point 31: Line 272: “Feature Map” shouldn’t be capitalized
Response 31:Thank you very much for your suggestion, we all agree with you. and revised in the manuscript.
Point 32: ##### NOTE: I will stop providing grammar and punctuation advice from this point as some very extensive revisions of these issues are required, and it falls outside the purpose of reviewing the scientific content. ##########
Response 32:Thank you very much for your questions, we have further checked the content of the manuscript.
Point 33: Figure 5: recommend making it vertical rather than horizontal
Response 33:Thanks a lot for your advice, but we're sticking to our original point of view. Because it really takes up a lot of space.
Point 34: Line 346: What is the P exactly?
Line 347: It’s not clear what the “AGi” term means.
Equation (9): what is “GT information”? Again, anchors need to be discussed.
Response 34:P means proposal, and the model will continuously update the value of (x, y, w, h, ). During this process, the model makes its recommendations, which it considers the most recent predicted value to be the closest to the true result. "GT information" represents the most ideal anchor box and the most realistic anchor box, which can perfectly fit the edge of the target and is the anchor box we will finally get. It includes (x, y, w, h, ) five parameters. “AGi” represents the data of the anchor box currently predicted by the model, because the model believes that the current anchor box data is closest to "GT", so we use “AGi” to represent the currently predicted result.
Point 35: Line 366: “the optimal weights” They are not optimal, but they might be more effective than random initialization.
Response 35:The "optimal weight" here refers to selecting the optimal weight from a series of operations of selection, crossover, and mutation from the generated multiple groups of initial weights. For example, he is the tallest in our class, and the tallest here refers to the tallest in our class, not in the world. It should be noted that even if the height is limited to our class, it is more effective than randomly selecting a person. The same is true for optimal weights.
Point 36: Line 371: “this paper proposes a convolutional neural network combined with the genetic algorithm to optimize the parameters to find the global optimal solution” This is an extremely high claim. The authors must prove it if they claim it’s true. I doubt it is true. It’s rather confusing this approach to initializing the net is being compared to gradient descent because almost all papers just randomly initialize with Gaussian or uniform random variables. And, then they discuss optimizing the network. It needs to be clarified as to whether they are talking about initialization or actual model fitting.
Response 36:Thank you very much for your question, the expression here is too radical and not rigorous enough. We have revised this sentence. Details are as follows:
"This paper proposes a convolutional neural network combined with a genetic algorithm to optimize parameters to be as close to the global optimum as possible". Almost all papers use stochastic gradient descent as their optimization network. Random initialization necessarily has its limitations. It is worth emphasizing that we have not completely abandoned the current optimization method. We replace its random part with a genetic algorithm, and combine the genetic algorithm with gradient descent as the optimization network. At the same time, we also tested on the CIFAR10 dataset, using the existing optimization network SGD and Adam as the control group, and verified from two aspects of finding the optimal path and training results. What we improved is the part of initializing the hyperparameters, but the whole process needs to be combined with the actual model results, which is a dynamic process.
Point 37: Line 373: The precise genetic algorithm being used needs to be outlined, rather than just referencing a book on genetic algorithms.
Response 37:Thank you very much for your question, we do have some work on exact genetic algorithms, the process is very complicated. In order to make the title of the article more concise and clear, we briefly describe the population and objective function of the genetic algorithm, but do not give specific selection, crossover, mutation processes, nor detailed operations, and no trouble. Many genetic algorithm books have detailed instructions. Because the genetic algorithm was proposed by others, it is obviously unreasonable to write a lot of other people's things in the paper. However, we have also done detailed operations, but they are not introduced in the paper, as follows:
- a. Encoding: encoding length when encoding in binary.
(1)
Where (a, b) is the value range of the independent variable, eps is the required precision, and when there are multiple independent variables, the code length is the sum of the code lengths of each independent variable.
(2)
- b. Decoding: Convert binary numbers to decimal numbers.
(3)Where L is the encoding length, X is the binary data.
- Fitness function: find the minimum value
(4)
where is an appropriately large number, and f(x) is determined by formula (10) as the objective function.
- Scale transformation of fitness function: Here, the dynamic linear transformation method is selected to search for the optimal solution, and the function expression is:
(5)
where M represents the total number of individuals in the population, and r is a random number between [0.9, 0.999].
- Selection operator: deterministic sampling selection method.
â‘ Calculate the fitness value of each individual â‘¡The probability of each individual being selected is:
(6)
represents the fitness value of each individual, the survival expectation value of each individual in the next generation.
(7)
M is the number of individuals in the population, representing the fitness value of each individual
â‘¢ Calculate the total number of people added to the next generation from the expected number of survival of each individual.
(8)
M is the number of individuals in the population, and [] represents the integer part of the expected survival value of individuals.
- Crossover: choose three-point crossover, assume that the length of the binary gene sequence is n, and perform 0, 1 exchange at the , , and nth positions of each gene sequence respectively.
- Mutation: The mutation probability is specified as 0.03, assuming that the length of the binary gene sequence is n, a random number is generated in [1, n] to represent the position of the mutation sequence, and a random number r is generated between [0, 1], if r <, then 0 becomes 1, and 1 becomes 0. When all the individuals of a certain generation of population are concentrated together, it is easy to cause the phenomenon of premature maturity and fall into the local optimal solution. In order to solve such problems, a large mutation operation is introduced. When there is a concentration phenomenon among individuals, the mutation probability of genes is increased, thereby increasing the probability of individuals jumping to other dimensions. When the condition is satisfied, the large mutation operation is performed, and is the density factor, which represents the degree of individual concentration.
Point 38: Line 378: again, it’s almost certainly not optimal.
Response 38:Thank you very much for your suggestion, we have made the following changes:
Point 39: Line 383: What fitness function is being used? How do we measure how good initial parameters are before we run gradient descent?
Response 39:Both the fitness function and the objective function being used are Equation 9, and they can be the same function. The quality of the initial hyperfunction can only be judged by the final Loss value, which cannot be measured before running gradient descent. However, the initial value of our model, also after each round of training, selects the smallest Loss. The current hyperparameter, its value is used as the initial test value for the next round, which is a process of repeated cycles.
Point 40: Figure 8: the word “optimal” is just used incorrectly throughout the paper
Response 40:Thank you very much for your question, we originally wanted to express finding the optimal path, it's a process, not a result. Obviously, the current description is wrong, and the specific modifications are as follows:
Simulation Experiment
Point 41: Lines 462-464: spacing and punctuation is inconsistent
Response 41:Thank you very much for your question, we have made changes.
Point 42: Lines 473-474: what do you mean by “converting the labeled dataset into COCO”? Probably the COCO format is meant here?
Response 42:The COCO dataset is a large-scale dataset that can be used for image detection, semantic segmentation, and image caption generation. The COCO dataset has different annotation types. These annotation types are stored in "JSON" format. In many model frameworks, the form of COCO dataset is often required. Our model also uses COCO dataset, and COCO dataset annotation is also very convenient.
Point 43: Line 475: “the initial parameters of the network layer” Which layer? Is pretraining not used?
Response 43:This refers to the hyperparameters of the network, which does not represent a certain layer. There is an error in the expression here, and we have made changes as follows: the hyperparameters of the model.
Point 44: Line 476, Table 2: these are hyperparameters. Parameters are the values the model learns. It seems genetic algorithm is probably used to search the hyperparameter space, in which case, it definitely 3does not find optimal hyperparameters.
Response 44:Thank you very much for your question, we've put it in a more moderate way.
Point 45: Table 1: GTX 3080Ti doesn’t exist – it’s RTX, PyTorch is not a “web framework”, there’s no spacing with Anaconda, and CUDA 1.0 is not correct. (It’s probably 10.0 or 11.0.)
Line 493: spacing
Line 509, 512: spacing/punctuation
Response 45:Thank you very much for your question, which we have revised in our latest manuscript.
Point 46: Lines 520-522: is there a train/test
Lines 523-524: “method proposed in this paper can achieve pixel-level segmentation for the fault location” What is the performance on pixel-level segmentation? Only recognition performance is mentioned.
Response 46:Thank you very much for your question. When evaluating the recognition effect of a model, the test effect is often given, and the training result does not indicate its generalization. So here is the test performance. Since the training performance is not convincing enough, it is unnecessary to express it again.
The pixel-level segmentation here is embodied in the rotated RoI Align. Abandon the rounding process when obtaining the position information of the bounding box. The binomial interpolation method accurately calculates the coordinates of the four corners of the frame, and has a higher number of pixels to complete the splitting task. However, it is difficult for us to find a metric to measure its performance, which we need to discuss in future work.
Point 47: Line 532: It should be written YOLOv3 (some other instances of this issue appear in the paper)
Line 550: these performance metrics should not be referred to as parameters since this word has a specific meaning in the deep learning context (this reappears multiple times)
Lines 557-558: What is S? It doesn’t appear above.
Response 47:Thank you very much for your question, we have made changes to address the above.
We have revised the issue of "YOLOv3" throughout.
We modify the parameters here as factors.
S is redundant, it originally belonged to a formula, and we modified it so that S does not have to appear.
Point 48: Line 562: how is FPS used by TOPSIS? It doesn’t seem to appear in the math. Why are there no segmentation or bounding box performance metrics? These are important beyond simply classification performance.
What is the relevance of reference [38]?
Response 48:The full name of TOPSIS is Technique for Order Preference by Similarity to an Ideal Solution. It ranks the limited number of evaluation objects according to how close they are to the idealized goal. It is to evaluate the relative merits and demerits of existing objects. It is a commonly used and effective method in multi-objective decision analysis, also known as the distance method between superior and inferior solutions. FPS and Accuracy are important indicators of the model. One is the image recognition speed, and the other is the recognition accuracy. Existing fault detection models have their own limitations. Some of them are fast but low in accuracy, and some are high in accuracy but slow. It is difficult to directly judge which is better. TOPSIS can help us solve this problem. FPS reflects the image processing time. Of course, the shorter the time, the better, and the Accuracy, of course, the bigger the better. Therefore, formula 17 is regularized, and the shorter the better time is converted into the bigger the better. Specifically, first, among the six models, the processing time of the respective model is subtracted from the longest processing time. Then, find out the respective maximum and minimum values under the FPS and Accuracy indicators. This part is completed by formulas 20-21. Formulas 22-23 respectively calculate the distance between each index of the model and the maximum and minimum values. Finally, a comprehensive score is given by TOPSIS. For the bounding box performance metric you mentioned, it is difficult for us to judge its performance with the size of the anchor box, because the size of the anchor box changes with the detection target. In future work, we also try to solve this problem. The TOPSIS indicator is indeed a bit difficult to understand, and we try to use reference [38] to help readers understand. We have made modifications to 562 as follows:
Point 49: Lines 578-581: are they test performance or training performance? Was there a validation set in hyperparameter tuning? What is the localization performance? Ignoring that makes use of object detectors and segmentation models somewhat pointless. Why are “precision” and “recall” capitalized?
Response 49:They are all test performance. We exclusively use the test set to further determine the hyperparameters in the model. This process is necessary and the generalization performance is good. For what you said about ignoring the use of object detection and segmentation models, our focus is on solving the problem. No matter what kind of model it is, as long as it can efficiently identify the type of fault and locate it accurately, it is a good model. "Precision" and "Recall" as proper nouns, almost all articles are capitalized, which can also be proved by referring to [20].
Point 50: Line 617: how are the boxplots computed? Are the models being run multiple times on random samples and accuracies measured? If so, how many times?
Response 50:In the test data, we randomly selected 500 test results. Plot a boxplot with these 500 results. Specifically, first arrange these test results (value of Accuracy) in ascending order, and then find the median, upper quartile, and lower quartile of this group of numbers. The upper and lower edges represent the maximum and minimum of these data. The height of the bins partly reflects how volatile the data is, and sometimes there are some "outliers", outside the bins.
Point 51: Line 629: These aren’t histograms
Response 51:Thank you very much for your question, we have made changes.We changed "Histograms" to "Bar Charts"
Point 52: Line 640: “smallest error” – does it mean the error bar? How are the upper and lower bounds computed?
Response 52:It's the error bars, upper and lower bounds = mean +/- 1.96SE. SE stands for Standard Error (SE).
Point 53: Discussions and Future Work
Line 651: hyperparameters
Line 659: “the author himself”
Are there not three authors?
References
Reference 6: “Proceedings of …” should be italic rather than the article title
Response 53:Thank you very much for your question, we have made changes to address the above.
In response to the problem of inconsistent character spacing in manuscripts, this is because if the last word of each line cannot be written, it needs to be wrapped. Adding hyphens can cause other problems. Regarding the character spacing, the editor will help us deal with it at the end. Finally, we thank you again for your questions, which have greatly improved the quality of the manuscript.

Reviewer 2 Report
Authors should address the following comments in the revision.
1) The abbreviation at the first used should be defined. RCNN, GA-DA,...
2) "The current fault diagnosis methods can be divided into two camps, one is the physical method, and the other is themethod38 based on deep learning."- add reference for this statement, I suggest the following: (A New Statistical Features Based Approach for Bearing Fault Diagnosis Using Vibration Signals)
3) "As a traditional diagnostic method, physical methods have the advantages of real-time and high precision, mainly including ultrasonic, ultraviolet pulse, terahertz, and other methods"- add reference for this statement.
4) "In recent years, with the continuous development of artificial intelligence technology, detection methods based on deep learning frameworks have beenwidely60 used. ", add reference for this statement.
5) The summary of related work should be added at the end of introduction section. Moreover, add the major contributions and manuscript organization.
6) The dark sides of this work should be defined under the conclusion section.
Author Response
Response to Reviewer 2 Comments
We are very grateful to the editors and reviewers for your valuable comments on our manuscript, which has greatly helped to improve the quality of our manuscript. We re-edited various parts of the manuscript based on the reviewer’s comments. Below I will give a point-to-point response to each review.
Point 1:The abbreviation at the first used should be defined. RCNN, GA-DA,...
Response 1:We greatly appreciate your pointing out issues in the manuscript, which are valuable. We do ignore the definition of proper nouns. To give interested readers an in-depth understanding of the content of the article, we have included explanations of relevant acronyms in this manuscript. The specific modifications are as follows:
In line 11 of the manuscript, we modify "Mask RCNN" to "Mask of the regions with CNN features(Mask RCNN)".
"ARG-Mask RCNN" on line 13 is modified to "Mask RCNN of Attention, Rotation, Genetic algorithm (ARG-Mask RCNN)".
"ResNet101" in line 15 is modified to "Residual network 101(ResNet101)".
"GA-GD" on line 21 is modified to "Genetic Algorithm Combined with Gradient Descent (GA-GD)".
"Cascade RCNN" on line 67 is modified to "Cascade of the regions with cnn features(Cascade RCNN)".
"SSD" on line 68 is modified to "Single shot multibox detector (SSD)".
"YOLO" on line 69 is modified to "You only look once(YOLO)".
"RRoI Align" on line 364 is modified to "Rotated region of Interest Alogn(RRoI Align)".
"Adam and SGD" on line 437 is modified to "Adaptive moment estimation(Adam) and stochastic gradient descent(SGD)".
Point 2: "The current fault diagnosis methods can be divided into two camps, one is the physical method, and the other is themethod38 based on deep learning."- add reference for this statement, I suggest the following: (A New Statistical Features Based Approach for Bearing Fault Diagnosis Using Vibration Signals).
Response 2:We really appreciate your helpful advice. We agree that adding (A New Statistical Features Based Approach for Bearing Fault Diagnosis Using Vibration Signals) references makes the article more convincing. The content added in the specific article is as follows:
The current fault diagnosis methods[8] can be divided into two camps, one is the physical method, and the other is the method based on deep learning.
- 8. Altaf, M.; Akram, T.; Khan, M. A.; Iqbal, M.; Ch, M. M. I.; Hsu, C.-H., A New Statistical Features Based Approach for Bearing Fault Diagnosis Using Vibration Signals. Sensors 2022,22, (5), 2012.
Point 3: "As a traditional diagnostic method, physical methods have the advantages of real-time and high precision, mainly including ultrasonic, ultraviolet pulse, terahertz, and other methods"- add reference for this statement.
Response 3:The question you raised gave us a lot of inspiration, and we found that the sentence (As a traditional diagnostic method, physical methods have the advantages of real-time and high precision, mainly including ultrasonic, ultraviolet pulse, terahertz, and other methods) did introduce a reference. If it is just a monotonous sentence without introducing references, I think it is difficult to convince readers to agree with our point of view. We unanimously decided to insert the relevant statement here. The specific modifications are as follows:
As a traditional diagnostic method, physical methods[9] have the advantages of real-time and high precision, mainly including ultrasonic, ultraviolet pulse, terahertz, and other methods.
- 9. Henao, H.; Capolino, G.-A.; Fernandez-Cabanas, M.; Filippetti, F.; Bruzzese, C.; Strangas, E.; Pusca, R.; Estima, J.; Riera-Guasp, M.; Hedayati-Kia, S., Trends in fault diagnosis for electrical machines: A review of diagnostic techniques. IEEE industrial electronics magazine 2014,8, (2), 31-42.
Point 4:4) "In recent years, with the continuous development of artificial intelligence technology, detection methods based on deep learning frameworks have beenwidely60 used. ", add reference for this statement.
Response 4:Thank you very much for your suggestion, we all agree with you. (In recent years, with the continuous development of artificial intelligence technology, detection methods based on deep learning frameworks have been widely used.) Do need to add a reference here. To demonstrate mastery of cutting-edge work, we need to include recently published work. This also helps readers better understand the content and novelty of the work, making the article more persuasive.The specific modifications are as follows:
In recent years, with the continuous development of artificial intelligence technology, detection methods based on deep learning frameworks have been widely used[13].
- Tong, K.; Wu, Y.; Zhou, F., Recent advances in small object detection based on deep learning: A review. Image and Vision Computing 2020,97, 103910.
Point 5:The summary of related work should be added at the end of introduction section. Moreover, add the major contributions and manuscript organization.
Response 5:Thank you so much for such meticulous advice. In order to more clearly express our in-depth exploration of cutting-edge work and to clearly point out the contributions of this paper, we summarize related work in the Introduction section, while adding major contributions and manuscript organization. The specific additions are as follows:
In general, these existing advanced insulator fault diagnosis methods have their advantages, but some flaws are hard to hide. Physical methods such as ultrasound, ultraviolet pulse, and terahertz, it has the advantages of real-time and high precision. However, they also have common shortcomings, such as it is difficult to achieve large-area outdoor detection, and the efficiency is relatively low. For SSD, RetinaNet, YOLO, Cascade RCNN, and Mask RCNN, these deep learning-based methods have high efficiency and can meet the needs of large-scale outdoor detection, but they have contradictions in real-time and accuracy. Specifically, single-stage target detection algorithms such as SSD, RetinaNet, and YOLO have fast recognition speed, but low accuracy. The two-stage target detection algorithms such as Cascade RCNN and Mask RCNN are characterized by high accuracy, but slow speed and difficult to realize real-time monitoring of insulators. It is worth noting that these deep learning-based methods only detect a single fault type, and they cannot complete the multi-fault classification task.
To complete the detection of various faults under the premise of real-time and high precision. In this paper, a fault diagnosis method for infrared insulators based on ARG-Mask RCNN is proposed. First, it is proposed to modify the 7×7 convolution kernel of the first layer of the backbone network ResNet101 to a three-layer 3×3 convolution kernel. The three-layer 3×3 convolution kernel has the same receptive field as the 7×7 large convolution kernel. However, the amount of computation is much smaller than that of the large convolution kernel, and an attention mechanism is added to reduce the amount of network computation and improve the detection speed of small targets. Subsequently, a rotation mechanism is added to the calculation formula of the improved loss function to improve the positioning accuracy of the target insulator and effectively separate the target from the background. After that, it is proposed to improve the initial parameters, and the updated parameters originally generated by Mask RCNN are now generated by a genetic algorithm, to obtain the global optimal solution and improve the identification accuracy of faulty insulators. Then, the labeled dataset is trained to analyze various misdiagnosis phenomena and their causes in the detection results. Finally, the ARG-Mask RCNN method proposed in this paper has obvious advantages through application experiments and comparative analysis. This research has the following contributions:
(1) A new backbone network is proposed to improve the capability of fault feature extraction.
(2) A rotating anchor frame is proposed to make the segmentation of fault locations more accurate.
(3) The genetic algorithm combined with the gradient descent method is proposed to optimize the parameters so that the model is as close to the global optimal solution as possible, and the detection accuracy of the model is improved.
(4) By comparing with several optimal insulator fault identification algorithms, the superiority of the proposed method is confirmed.
The rest of this article is organized this way. Section 2 briefly introduces the four most common insulator faults and the Mask RCNN base network. Section 3 introduces the ARG-Mask RCNN network in detail from three aspects: backbone network, loss function, and parameter optimization. Section 4 mainly demonstrates the superiority of this method in practical detection. The conclusion is in Section 5.
Point 6: The dark sides of this work should be defined under the conclusion section.
Response 6:We appreciate your pointing out the problems in the manuscript, and we've been candid about the dark side of this work. Your suggestion has helped us a lot. It is not only reflected in improving the quality of the article, but also in seeking truth from facts, which will always be with us in future work. We have indeed overlooked the definition of inadequacy. The specific additions are as follows:
Many factors cause the failure of insulators, but most of them are determined by natural factors. What we can do is to find it as soon as possible and reduce unnecessary losses. Deep learning methods are a popular method in the field of insulator fault identification. Although the method proposed in this paper has achieved good results, there are still some dark sides worthy of further study: (1) In the actual fault detection, the influence of various bad weather should be considered. For example, in the background of rainy and dense fog, the detection accuracy of the model will drop slightly. (2) There are slight differences between some faults, which cause the network to fail to identify such faults normally, and even confuse faults with similar characteristics. (3) There are many kinds of faults of insulators. This paper only covers four common fault detections: self-explosion fault, contamination fault, zero fault, and damage fault. For some uncommon types of failures, it is not yet possible to identify them.

Round 2
Reviewer 1 Report
The revised version of the paper is much better in many ways.
It is communicated why a segmentation model is being used, but the segmentation performance is never evaluated, so the following claims aren't justified by the authors:
Line 138: "A rotating anchor frame is proposed to make the segmentation of fault locations more accurate"
Line 562: "It can be concluded that the method proposed in this paper can achieve pixel-level segmentation for the fault location, and can accurately identify different fault types, which greatly consolidates the safe and stable operation of the power grid"
Some performance measures for segmentation performance should be included or these claims should be removed.
Some additional issues:
1. The framerates are low across methods if an RTX 3080Ti is being used, as reported. YOLOv3--even the full version--runs at 37 fps in the YOLOv3 paper on a GPU slower than RTX 3080Ti.
2. Prior replies indicate use of the test set in hyperparameter tuning when it should not be viewed during tuning.
Author Response
Response to Reviewer 1 Comments
We again thank Reviewer 1 for his valuable comments on our manuscript, which have been of great help in improving the quality of our manuscript. We have re-edited various parts of the manuscript based on the comments of the reviewers. Below I'll give a point-to-point response to each comment.
Point 1:It is communicated why a segmentation model is being used, but the segmentation performance is never evaluated, so the following claims aren't justified by the authors:
Line 138: "A rotating anchor frame is proposed to make the segmentation of fault locations more accurate"
Line 562: "It can be concluded that the method proposed in this paper can achieve pixel-level segmentation for the fault location, and can accurately identify different fault types, which greatly consolidates the safe and stable operation of the power grid"
Some performance measures for segmentation performance should be included or these claims should be removed.
Response 1:Thank you very much for your questions about our manuscript, it will go a long way in improving the quality of the manuscript. For the segmentation index, it is still unable to give a reasonable evaluation standard. As we all know, we cannot judge its performance by the size of the segmented target area, because it is determined by the actual size of the target. Even for the same target, the mask area will be different due to the shooting angle and distance. However, this is also something we need to consider in our future work. For now, in order to ensure the rigor of the article, we have unanimously decided to delete the statements about lines 138 and 562. The specific modifications are as follows:
- A rotated anchor box is proposed to reduce the extraneous background in the prediction box.We show the effect of the rotation mechanism in the image below.
It can be concluded that the method proposed in this paper can identify a variety of fault types, which greatly consolidates the safe and stable operation of the power grid.
Some additional issues:
Point 2:The framerates are low across methods if an RTX 3080Ti is being used, as reported. YOLOv3--even the full version--runs at 37 fps in the YOLOv3 paper on a GPU slower than RTX 3080Ti.
Response 2:I have to say, your question is of a very high level. In many object detection tasks, the frame rate of YOLOv3 does reach 37FPS. However, the type of GPU is not the only factor contributing to low frame rates. For example, (1) the size of the input image. (2) Full use of GPU. (3) Resolution of the dataset. (4) Turn on multi-scale training (set random=1 in the detector). (5) The size of Batch_size, etc. We are using a high resolution dataset. Also, FPS may be low due to other computer configuration factors. As you said, all methods have low frame rates, which is true. However, the actual results are like this, and we respect the experimental results.
Point 3:Prior replies indicate use of the test set in hyperparameter tuning when it should not be viewed during tuning.
Response 3:Thank you very much for your question, there is indeed a bug here. In deep learning, hyperparameters are values that are set before learning begins, not the data the model is trained on. Also, we can only tune hyperparameters based on training results (loss values). For the adjustment of hyperparameters, we use the self-made "FAIN_detection" dataset, which is an extended dataset based on the original training set through rotation, translation, and other operations. It is worth emphasizing that it is neither from the training set nor from the test set. If the test set is used for hyperparameter tuning, it is cheating, and this behavior will not be tolerated. However, the statement here is indeed false, the dataset that is useful in hyperparameter tuning is independent of the test set. We modified this.

Round 3
Reviewer 1 Report
I recommend accepting the paper in the present form.
As with all articles, I recommend taking a little time to proofread it once again. I always find mistakes in my own papers while making one final pass.